# An electrodiffusive network model with multicompartmental neurons and synaptic connections

**Marte J. Sætra** [1] *, **Yoichiro Mori** [2,3]

**1** Department of Numerical Analysis and Scientific Computing, Simula Research Laboratory, Oslo, Norway, **2** Department of Mathematics, University of Pennsylvania, Philadelphia, Pennsylvania, United States of America, **3** Department of Biology, University of Pennsylvania, Philadelphia, Pennsylvania, United States of America

* martejulie@simula.no

**Data Availability Statement:** The code described in the paper is freely available online at https://github.com/martejulie/electrodiffusive-network-model.

## Abstract

Most computational models of neurons assume constant ion concentrations, disregarding the effects of changing ion concentrations on neuronal activity. Among the models that do incorporate ion concentration dynamics, simplifications are often made that sacrifice biophysical consistency, such as neglecting the effects of ionic diffusion on electrical potentials or the effects of electric drift on ion concentrations. A subset of models with ion concentration dynamics, often referred to as electrodiffusive models, account for ion concentration dynamics in a way that ensures a biophysical consistent relationship between ion concentrations, electric charge, and electrical potentials. These models include compartmental single-cell models, geometrically explicit models, and domain-type models, but none that model neuronal network dynamics. To address this gap, we present an electrodiffusive network model with multicompartmental neurons and synaptic connections, which we believe is the first compartmentalized network model to account for intra- and extracellular ion concentration dynamics in a biophysically consistent way. The model comprises an arbitrary number of "units," each divided into three domains representing a neuron, glia, and extracellular space. Each domain is further subdivided into a somatic and dendritic layer. Unlike conventional models which focus primarily on neuronal spiking patterns, our model predicts intra- and extracellular ion concentrations ($Na^+$, $K^+$, $Cl^-$, and $Ca^{2+}$), electrical potentials, and volume fractions. A unique feature of the model is that it captures ephaptic effects, both electric and ionic. In this paper, we show how this leads to interesting behavior in the network. First, we demonstrate how changing ion concentrations can affect the synaptic strengths. Then, we show how ionic ephaptic coupling can lead to spontaneous firing in neurons that do not receive any synaptic or external input. Lastly, we explore the effects of having glia in the network and demonstrate how a strongly coupled glial syncytium can prevent neuronal depolarization blocks.

**Funding:** MJS was supported by the Research Council of Norway via FRIPRO grant no. 324239 (EMIx), the U.S.-Norway Fulbright Foundation, and Epilepsiforbundets forskningsfond (Norway's Epilepsy Foundation). YM was supported by the University of Pennsylvania Materials Research Science and Engineering Center (MRSEC) grant DMR-2309043, the Simons Foundation Math+X grant 234606, and by the National Science Foundation through grant DMS-2042144. The funders had no role in study design, data collection and analysis, decision to publish, or preparation of the manuscript.

**Competing interests:** The authors have declared that no competing interests exist.

## Author summary

Neurons communicate using electrical signals called action potentials. To create these signals, sodium ions must flow into the cells and potassium ions must flow out. This transmembrane flow requires a concentration difference across the neuronal membrane, which the brain works continuously to maintain. When scientists build mathematical models of neurons, they often apply the simplifying assumption that these ion concentration differences remain constant over time. This assumption works well for many scenarios, but not all. For instance, during events like stroke or epilepsy, the ion concentrations can change dramatically, affecting how neurons behave. Moreover, recent literature suggests that changing ion concentrations also play an important role in normal brain function. To study these scenarios, we need models that can dynamically track changes in ion concentrations. The neuroscience community currently lacks a computational model describing the effects of ion concentration dynamics on neuronal networks, while maintaining a biophysical consistent relationship between ion concentrations and electrical potentials. To address the need for such a model, we have developed a neuronal network model that predicts changes in both intra- and extracellular ion concentrations, electrical potentials, and volumes in a biophysically consistent way.

## Introduction

Most computational models of neurons assume that the concentrations of the main charge carriers in the brain remain constant over time [1]. One can largely justify this assumption due to the small amount of ions needed to cross the membrane to generate an action potential (box 2.3 in [2]). Additionally, the brain has inherent mechanisms, such as pumps and cotransporters, that work to keep ion concentrations near baseline levels even in the face of numerous action potentials. Another assumption commonly made by modelers is that the extracellular potential is spatially uniform and grounded ($\phi_e = 0$). This can be a useful approximation as the extracellular potential is typically so small that one can neglect its ephaptic effects on neurons without severe loss in accuracy [1, 3]. However, the assumption of constant ion concentrations and a grounded extracellular potential does not always hold. It has long been recognized that ion concentrations may shift in association with various pathologies, such as seizures, stroke, or spreading depression (a slow wave of hyperactivity followed by neuronal silence associated with, i.a., migraine aura [4, 5]) [6, 7]. Such changes are sometimes accompanied by changes in the extracellular potential that can reach several tens of millivolts in magnitude [1, 8, 9]. Moreover, there is a growing recognition that changes in ion concentrations may play an active role in normal brain function. For instance, Ding et al. [10] discovered small changes in the extracellular ionic environment ($K^+$, $Ca^{2+}$, and $Mg^{2+}$) in mice accompanying the transition between sleep and wakefulness. To investigate these kinds of scenarios, we need models that account for ion concentration dynamics and their effect on neuronal function.

Several existing models account for ion concentration dynamics, see e.g. [1, 11–40]. A particular subset of these models, sometimes referred to as electrodiffusive neuron models, consider both diffusion and electric drift in the transport of ions [1, 24–33]. Additionally, these models consider non-zero, spatially non-uniform extracellular potentials. The models emphasize biophysical consistency, ensuring that all ionic movement affects concentrations and electrical potentials, and vice versa [41]. One approach to electrodiffusive modeling is to utilize

multicompartmental neuron models. We have previously published the first multicompartmental neuron model that accounts for intra- and extracellular ion concentration dynamics in a biophysically consistent way [1]. Later, we introduced the neuron-extracellular-glia model (edNEG), which extends the initial model by including volume dynamics and a glial domain [28]. While these models are useful for exploring the interplay between ion concentration dynamics and neuronal activity, they are limited to studying a single cell or a tiny piece of tissue. For example, these models are not suitable for studying the propagation of spreading depression. Another approach to electrodiffusive modeling is to consider explicit geometries [24–27]. These types of models are superior when it comes to geometrical detail and to uncover relationships between shape and function. The models can in theory be expanded to include an arbitrary number of cells, but the computational cost of simulating hundreds of neurons is prohibitively expensive. A final approach to electrodiffusive modeling is to utilize domain-type models [29–33]. In this type of framework, the model represents an average piece of brain tissue, and each point in the domain is interpreted as a certain average. The domain-type models are computationally more efficient than the geometrically explicit models and useful for studying phenomena such as spreading depression. However, due to their homogenized nature, it is not possible to track the activity of individual neurons within the tissue using these models.

With single-cell compartmental models, geometrically explicit models, and domain-type tissue models already in the toolbox, the neuroscience community currently lacks an electrodiffusive network model that allows tracking of individual, compartmentalized cells. Although a few models exist that account for ion concentration dynamics in neuronal networks to some extent, these models make simplifications that compromise biophysical consistency [34–40]. These simplifications are made by ignoring the combined effect of diffusion and electric drift on ionic transport, or neglecting the transport of ions through the extracellular space altogether. Some may not take all membrane currents into account when calculating ion concentrations, and/or they set some of the ion concentrations constant.

We envision that an electrodiffusive network model would be useful for studying not only how changes in ion concentrations affect various pathologies at the network level, but also, as recent literature suggests, how ion concentration dynamics influence healthy brain function. To address the need of such a model, this paper introduces an electrodiffusive network model with multicompartmental neurons and synaptic connections. The model is, to our knowledge, the first compartmentalized network model that describes intra- and extracellular ion concentration dynamics in a way that ensures biophysical consistency between ion concentrations and electrical potentials. The model builds on our previously published edNEG model and describes an arbitrary number of two-compartment neurons connected in a 1D arrangement, along with extracellular space and a glial syncytium. In the first part of the paper, we introduce the model and detail its features. We then showcase how ion concentration dynamics can give rise to behavior not captured by conventional models, including altered synaptic currents and spontaneous neuronal activity due to ephaptic coupling. We also show how glial $K^+$ buffering can prevent neuronal depolarization blocks.

## Results

### An electrodiffusive network model

The electrodiffusive network model comprises an arbitrary number of "units," each representing a neuron and its immediate surroundings (Fig 1). The units are divided into two layers: a soma layer (bottom) and a dendrite layer (top). These layers are further subdivided into three domains, resulting in a total of six compartments within each unit. Specifically, two

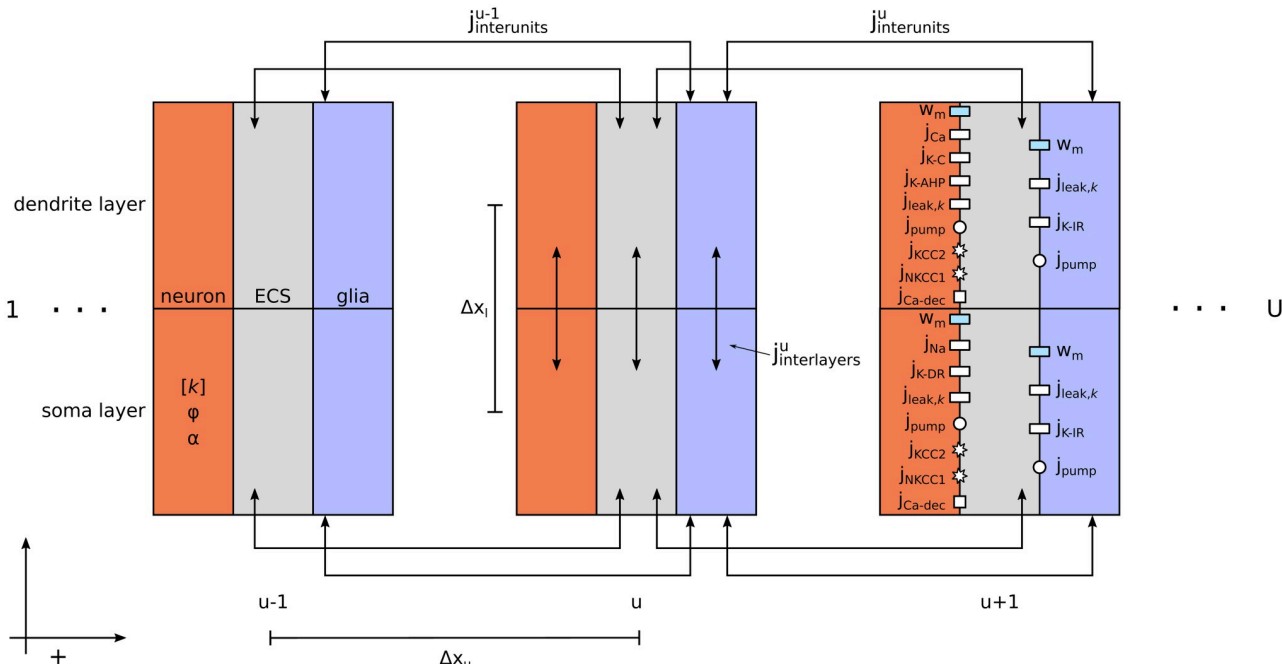

**Fig 1. Model schematics.** The electrodiffusive network model comprises an arbitrary number of "units" $U$, each divided into three domains representing a neuron, the extracellular space (ECS), and a glial segment. Each domain is further subdivided into two compartments representing somatic (bottom) and dendritic (top) layers, separated by a distance $\Delta x_l$. Utilizing the KNP framework, the model predicts the temporal evolution of the ion concentrations $[k]$ ($k = \{Na^+, K^+, Cl^-, Ca^{2+}\}$), electrical potential $\phi$, and volume fraction $\alpha$ in each compartment. Ionic movement is driven by diffusion and electric drift between layers ($j_{interlayers}$). Additionally, ions can move between units via the ECS or through the glial syncytium ($j_{interunits}$). Neurons are not coupled with eachother by electrodiffusion, but may communicate through chemical synapses (not illustrated). The membrane mechanisms are taken from our previously published edNEG model [28]. Specifically, both neuronal membranes feature ion specific leak channels ($j_{leak,k}$), a $Na^+/K^+$ pump ($j_{pump}$), KCC2 ($j_{KCC2}$) and NKCC1 ($j_{NKCC1}$) cotransporters, and a $Na^+/Ca^{2+}$ exchanger ($j_{Ca-dec}$). Additionally, the neuronal soma contains $Na^+$ ($j_{Na}$) and $K^+$ delayed rectifier ($j_{K-DR}$) channels, while the neuronal dendrite feature a $Ca^{2+}$ channel ($j_{Ca}$), a $K^+$ afterhyperpolarization channel ($j_{K-AHP}$), and a $Ca^{2+}$-dependent $K^+$ channel ($j_{K-C}$). Both glial compartments include ion specific leak channels ($j_{leak,k}$), an inward rectifying $K^+$ channel ($j_{K-IR}$), and a $Na^+/K^+$ pump ($j_{pump}$). The model also accounts for transmembrane fluid flow ($w_m$) driven by osmotic pressure gradients. The units can be arranged either linearly (i.e., with closed boundary conditions) or in a ring (i.e., with periodic boundary conditions), with a separation distance $\Delta x_u$ between them. Positive directions are defined from soma to dendrite, from intracellular to extracellular, and from left to right.

compartments are dedicated to representing the neuron, another two represent glial cells (i.e., astrocytes), and the remaining two represent extracellular space (ECS). Each unit can be interpreted in one of two ways, depending on parametric values [28]. It could represent a single neuron and the average piece of ECS and glial ion uptake available to that neuron. Alternatively, it could represent an average piece of tissue, comprising multiple neurons grouped together along with their shared ECS and glial ion uptake. In this paper, we treat each neuronal domain as representing a single neuron and its characteristics within the somatic and dendritic layers. The model predicts the temporal changes in ion concentrations ($Na^+$, $K^+$, $Cl^-$, and $Ca^{2+}$), electrical potentials, and volume fractions within each compartment, modeled using coupled ordinary differential equations (ODEs). The model is implemented using the Kirchhoff-Nernst-Planck (KNP) framework [41], ensuring charge conservation and biophysical consistency between ion concentrations and electrical potentials.

The units can be arranged either linearly, i.e., with closed boundary conditions, or in a ring, i.e., with periodic boundary conditions. For the neurons, ions can move intracellularly between the somatic and dendritic layers but not between different units without traversing the ECS. In the ECS, ions can move both between layers and different units. Interunit communication occurs between nearest neighbors. We assume that the glial domain represents an

astrocyte syncytium connected through gap junctions, allowing ions to move in the same manner as in the ECS but with different tortuosity value. All ionic transport between layers and units is driven by diffusion and electric drift.

Beyond ephaptic coupling, neurons have the ability to communicate via chemical synapses. This form of communication is represented by an action potential (AP) in the presynaptic neuron affecting the conductance of an ion channel within the membrane of the postsynaptic neuron. Notably, any neuron within the network has the potential to establish connections with any other neuron, irrespective of their relative positions.

Each unit in the model is adopted from our previously published edNEG model [28]. The neuronal and glial cells are equipped with cell- and ion-specific leak channels, cotransporters, and ion pumps whose activities are finely balanced to maintain ionic homeostasis. The neuronal membrane also incorporates active ion channels that differ between the soma and dendrite, enabling the generation of somatic APs and dendritic $Ca^{2+}$ spikes. Both glial compartments feature inward rectifying $K^+$ channels. The model accounts for cellular swelling driven by osmotic pressure gradients. A detailed description of the model is given in the Methods section.

## Electrodiffusive network dynamics: Electrical potentials, ion concentrations, and osmotic effects

While conventional network models primarily focus on neuronal spiking patterns, the electrodiffusive network model gives a comprehensive prediction of intra- and extracellular ion concentrations ($Na^+$, $K^+$, $Ca^{2+}$, and $Cl^-$), electrical potentials, and volume fractions. To showcase the model's capabilities and its distinction from conventional models, we conduct an initial simulation with ten units in a ring configuration, separated by a 5 μm distance (Fig 2A). We let the neurons be synaptically connected from soma to dendrite, with each neuron forming a connection with its right-hand neighbor. At $t = 0.1$ s, we let the first neuron ($u = 1$) receive a synaptic stimulus, mimicking a connection to a presynaptic neuron firing a single AP. The synaptic strength is set such that a single AP is sufficient to trigger an AP in the postsynaptic neuron under this particular simulation setup.

In the absence of external stimuli, all neurons maintain a resting membrane potential of −67 mV (Fig 2B). The synaptic stimulus initiates a cascade of APs through the network, with each neuron triggering the next in succession (Fig 2B). After approximately 0.15 s, the signal completes its propagation through the network, returning to the first neuron and starts a new cycle. Upon examining the neuronal membrane potentials more closely, we notice a combination of familiar and new behaviors compared to those observed in the edNEG model (Fig 3A, 3B and 3C). The edNEG model, which serves as the foundational structure of the network, features a characteristic AP shape [28]. When building the electrodiffusive network model, we managed to retain this shape by manually tuning the coupling strength between the layers (see Methods for the full list of parameters). The neurons exhibit new behavior when neighboring neurons fire APs. Specifically, when a neuron within the network fires, we see a subtle change in the membrane potentials of the other neurons (Fig 3C). This change in membrane potential is attributed to the neurons' electric ephaptic coupling, that is, electrical interactions through the ECS; When a neuron fires, it induces a change in the extracellular potential across all units, with an amplitude which depends on the distance to the spiking neuron (Fig 3D, 3E and 3F). This propagation of extracellular potential changes also manifests in the glial membrane potentials, which shift across all units each time a neuron fires, in a similar manner as the extracellular potentials (Fig 3G, 3H and 3I). On a slower time scale, the glial membrane

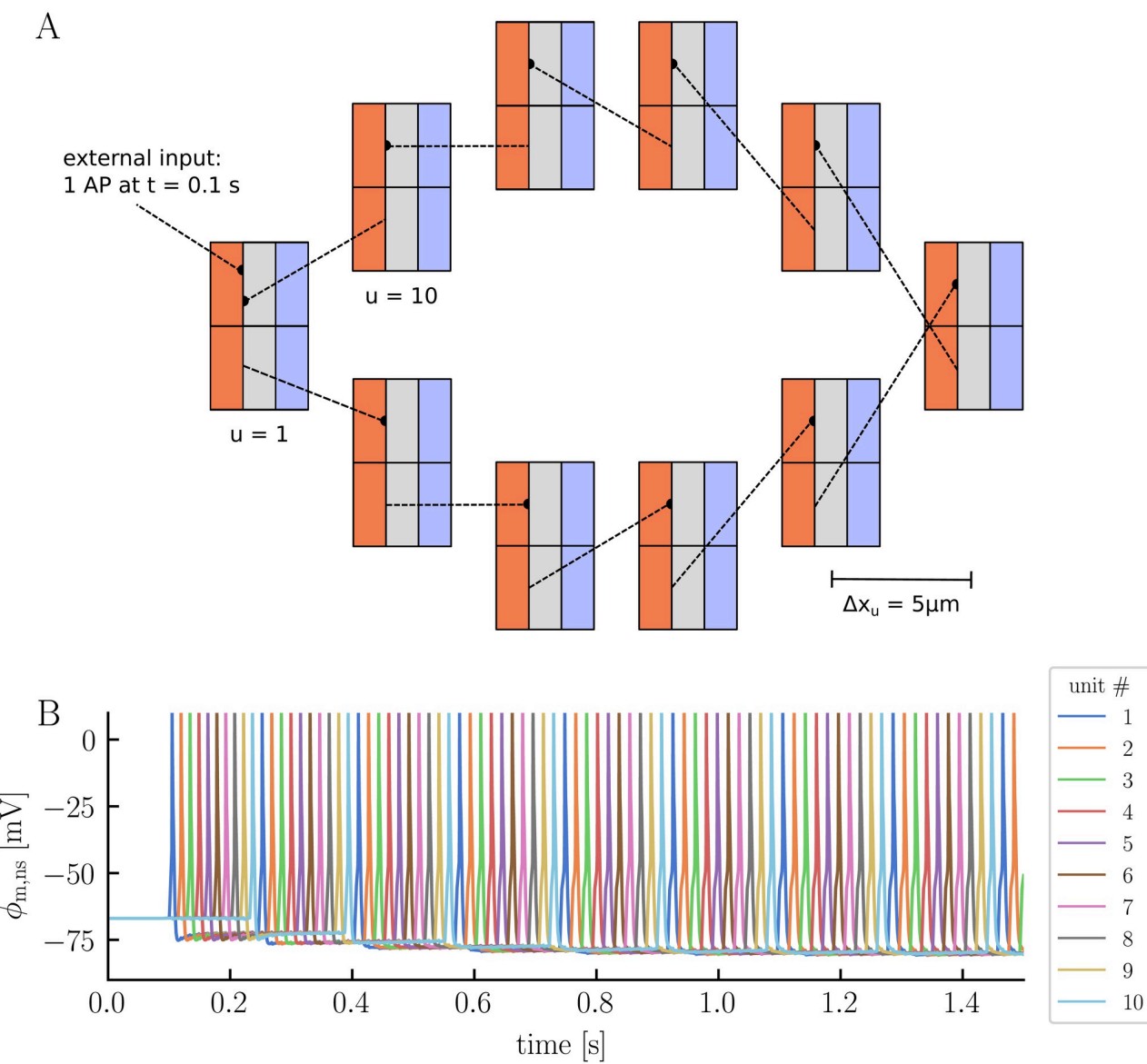

**Fig 2. The electrodiffusive network model in a ring configuration.** Panel **A** illustrates the simulation setup. Ten model units are placed in a ring configuration, separated by a 5 μm distance. The neurons are synaptically connected from soma to dendrite, with each neuron forming a connection with its right neighbor. Neuron 1 ($u = 1$) receives a single synaptic input at $t = 0.1$ s, making it fire an AP. All simulation parameters are listen in Methods. Panel **B** displays the neuronal somatic membrane potential across all units over time.

potential become gradually less polarized for all units, increasing from a resting membrane potential of −84 mV to −77 mV by the end of the simulation ($t = 1.5$ s).

To showcase how the ion concentrations in the system behave, we specifically examine the $K^+$ concentrations in the soma layer (Fig 4). Before stimulus onset, the intra- and extracellular $K^+$ concentrations remain at baseline values (Fig 4A, 4D and 4G). This equilibrium is maintainted by a delicate balance of leak currents, pumps, and cotransporters. Upon activation of the synaptic stimulus, the ECS $K^+$ concentration in the dendrite layer of unit 1 begins to rise, as this is where we apply the synaptic current (Fig 4C). When the neuron subsequently fires an AP, the ECS $K^+$ concentration in the soma layer follows, and we observe a rise of about 0.1

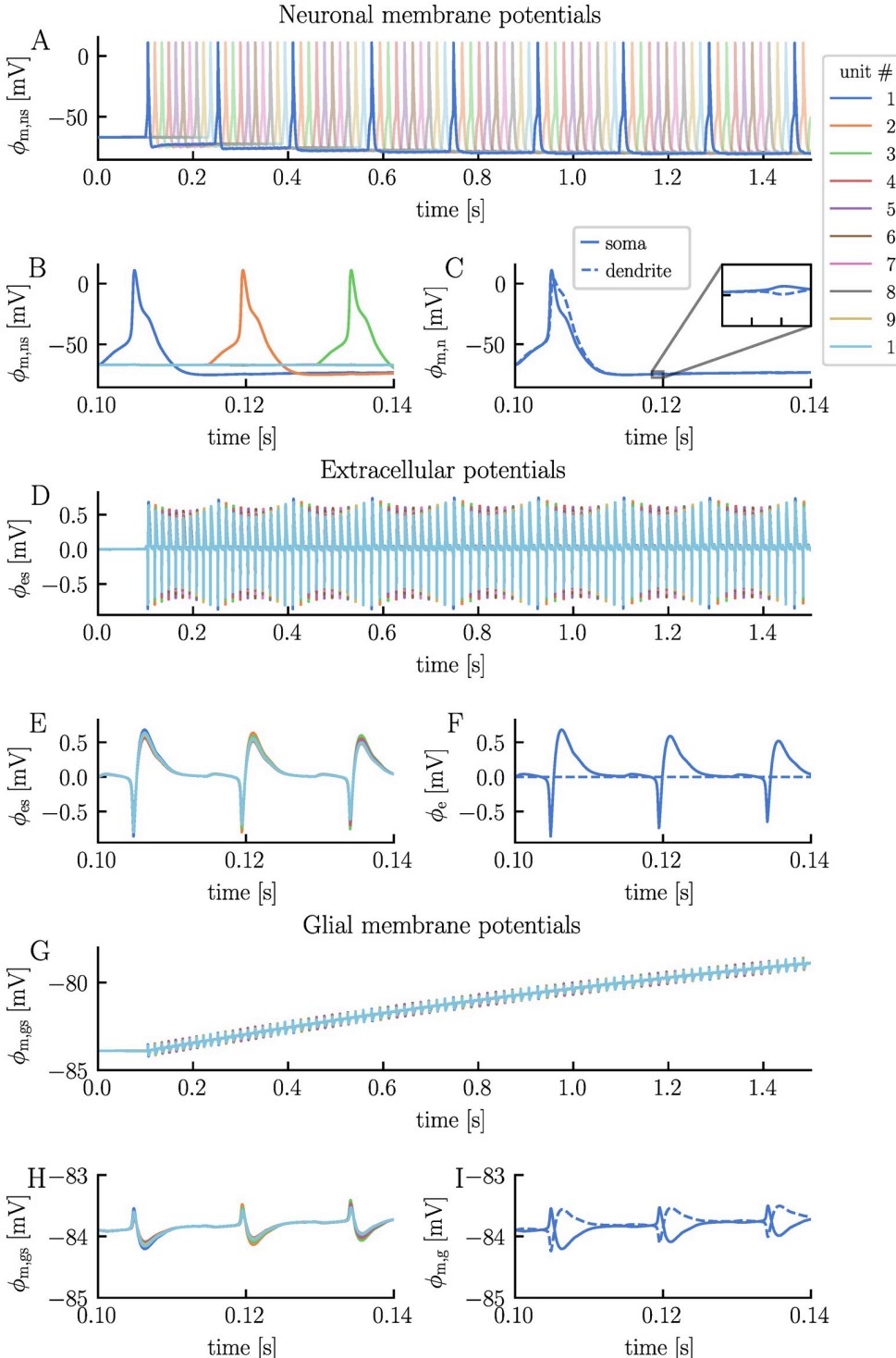

**Fig 3. Electrical potentials in the electrodiffusive network model.** The panels display the temporal evolution of various electrical potentials predicted by the electrodiffusive network model: the neuronal somatic membrane potential across units (**A** and **B**, with distinct x-axis scales), the somatic and dendritic membrane potential of neuron 1 (**C**), the extracellular potential in the soma layer across units (**D** and **E**), the extracellular potential in the somatic and dendritic layer of unit 1 (**F**), the glial somatic membrane potential across units (**G** and **H**), and the somatic and dendritic membrane potential of glia 1 (**I**). The extracellular potential of the dendrite layer in unit 1 is zero by definition and serves as reference point. The figure is based on the simulation presented in Fig 2.

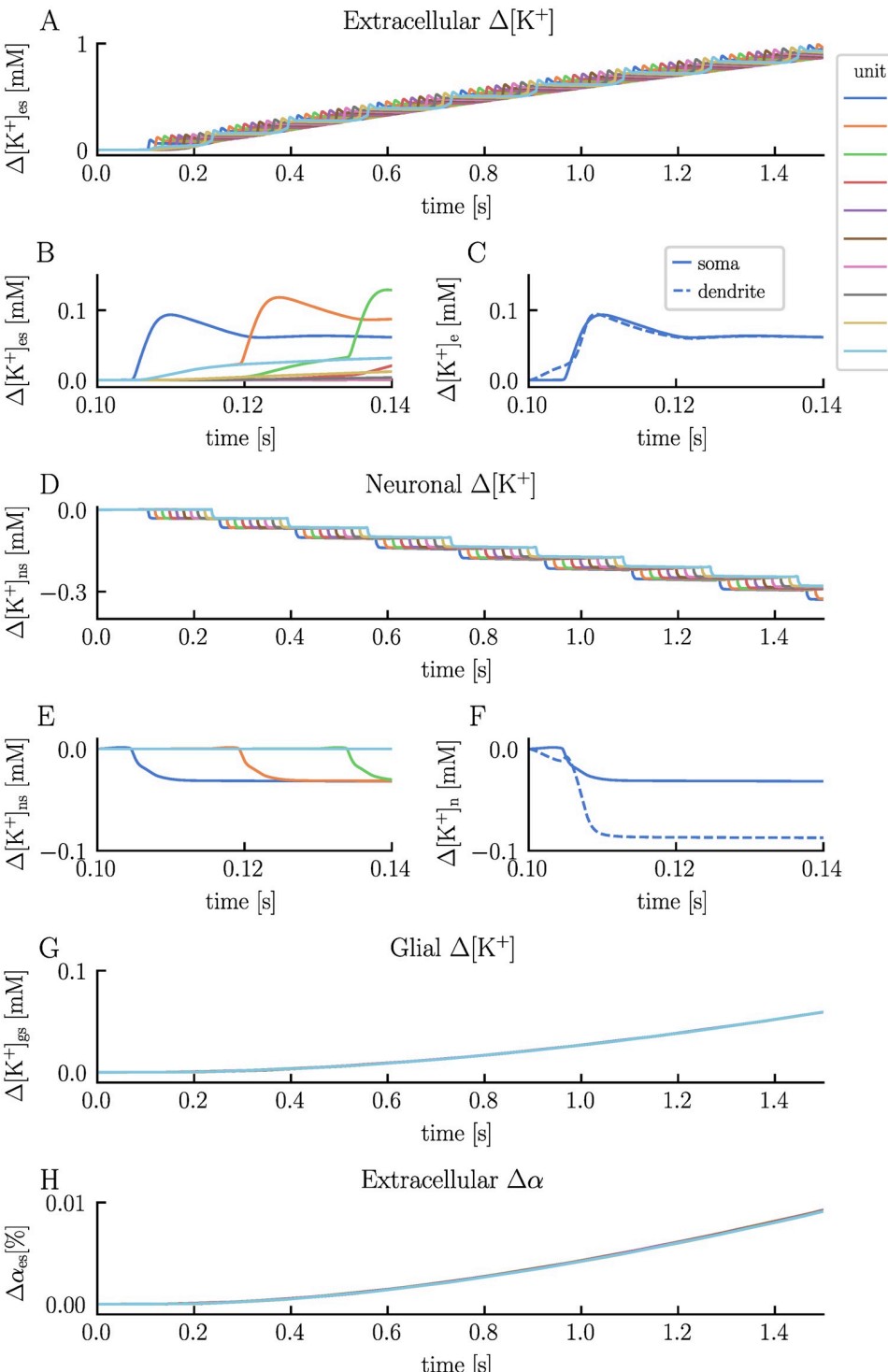

**Fig 4. Potassium concentrations and volume changes in the electrodiffusive network model.** The panels display the temporal evolution of various $K^+$ concentrations and volume fractions predicted by the electrodiffusive network model: the change in ECS $K^+$ concentration in the soma layer across units (**A** and **B**, with distinct x-axis scales), the change in ECS $K^+$ concentration in the somatic and dendritic layer in unit 1 (**C**), the change in neuronal somatic $K^+$ concentration across units (**D** and **E**), the change in somatic and dendritic neuronal $K^+$ concentration in unit 1 (**F**), the change in glial somatic $K^+$ concentration across units (**G**), and the change in ECS volume fraction in the soma layer across units (**H**). The figure is based on the simulation presented in Fig 2.

mM in both layers. After the AP ceases, these concentrations gradually decrease due to cellular uptake and diffusion into neighboring extracellular compartments. The diffusion of ECS $K^+$ from unit 1 into neighboring units is evident from the elevated ECS $K^+$ levels in unit 2 and 10, which starts to increase as soon as neuron 1 fires an AP (Fig 4B). Additionally, we notice a subtle increase in the ECS $K^+$ concentration in unit 3 and 9. The trend of decreasing ECS $K^+$ concentration in unit 1 persists until the next neuron fires an AP. For the neuronal $K^+$ concentrations, the picture is a bit different from the ECS (Fig 4D, 4E and 4F). Each time a neuron fires, the respective neuron experiences a decrease in the intracellular $K^+$ concentration. Since there are no gap junctions between the neurons, intraneuronal ions can only move across the neuronal membranes or between the layers. Consequently, the intraneuronal $K^+$ concentrations recover more slowly than in the ECS and stay close-to constant between each AP. However, over longer periods of inactivity, the intracellular ion concentrations return to baseline values together with the rest of the system (S1 Fig). In the glial compartments, we observe a steady increase in $K^+$ concentrations across all units due to $K^+$ uptake from the ECS (Fig 4G). Given the neurons' low firing rates and correspondingly minor increases in the ECS $K^+$ concentrations, the glial uptake of $K^+$ is moderate for this specific simulation. By the end of the simulation, the glial $K^+$ concentration has increased by about 0.06 mM across all units.

As the ion concentrations shift, there is a corresponding change in osmotic pressure gradients, inducing water flow and cellular volume changes (Fig 4H). Owing to the relatively brief duration of the simulation and the low firing rate within the current configuration, we detect only a slight change in the ECS volume, approximately 0.01% increase. However, in scenarios with considerably elevated firing rates, our model predicts shrinkage of the ECS, consistent with empirical observations (S2 Fig) [42, 43].

In summary, the electrodiffusive network model provides a broad range of outputs. These include intra- and extracellular ion concentrations, electrical potentials, and volume fractions, all intricately linked and represented in a biophysically consistent manner. We have already seen how incorporating extracellullar potentials results in electric ephaptic effects. In the following sections, we will delve further into the various features of the electrodiffusive network model.

## Elevated ECS $K^+$ concentrations strengthen synaptic currents

The electrodiffusive network model can easily accommodate both inhibitory and excitatory synapses, provided that the synaptic model(s) maintain ion conservation. Additionally, the electrodiffusive network model allows for synaptic connections in any spatial configuration. Ion conservation is ensured by designing ion-specific synaptic currents and incorporating them into the computation of ion concentrations. In this study, we implemented excitatory synapses that facilitate anion conduction, that is, synaptic activity induces post-synaptic transmembrane currents of $Na^+$, $K^+$, and $Ca^{2+}$. Changes in intra- and extracellular ion concentrations can dynamically influence these currents by altering the ionic reversal potentials. In this section, we will explore this effect in more detail.

To investigate the impact of ion concentration dynamics on synaptic currents, we conduct two simulations: one with the default modeling setup (full model) and another where the effects of ion concentration dynamics on neuronal activity are ignored (reduced model). In both simulations, we examine a single unit receiving a 400 Hz Poisson spike train between $t = 0.1$ s and $t = 10$ s (Fig 5A). Conseptually, this setup can be visualized as the neuron being surrounded by a group of other neurons that, on average, fire at the same frequency as the modeled neuron.

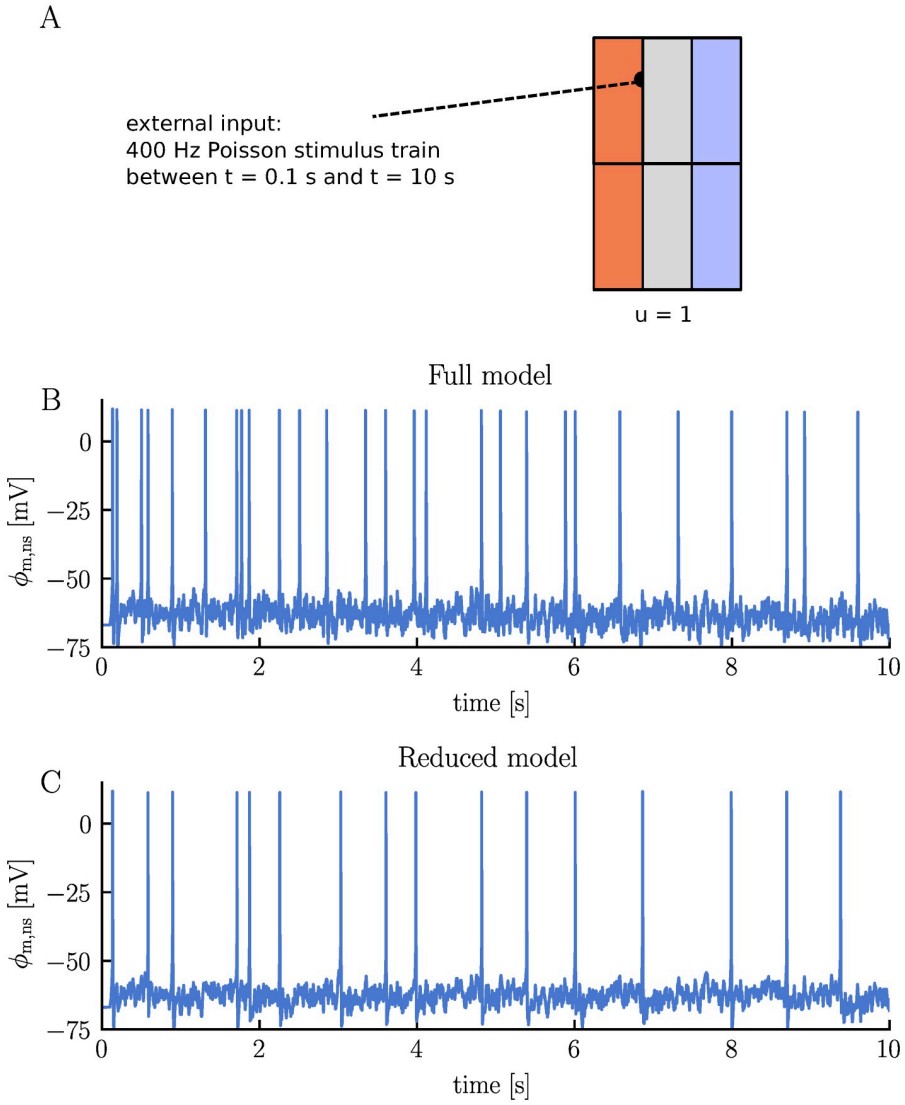

**Fig 5. Comparison of membrane potential dynamics in a single neuron undergoing synaptic stimulation: full vs. reduced model simulations.** Panel **A** illustrates the simulation setup. A single neuron is subjected to a 400 Hz Poisson stimulus train, delivered to the neuronal dendrite between $t = 0.1$ s and $t = 10$ s. We run two different simulations: one with the default modeling setup (Full model) and another ignoring ion concentration dynamics (Reduced model). All simulation details are given in Methods. Panels **B** and **C** display the neuronal somatic membrane potential over time for the full and reduced model, respectively.

The simulations reveal a notable difference in firing rates between the two models. The full-model neuron fires 27 APs over the course of 10 s, while the reduced-model neuron fires only 16 APs (Fig 5B and 5C, respectively). This difference in firing rates is attributed to different synaptic strengths. Upon examining the synaptic currents over time, we note that the current is, on average, stronger in the full model compared to the reduced model (Fig 6A and 6B, respectively). The models are subject to identical stimulus spike trains and have the same maximum conductances, so the different synaptic currents must stem from different ionic reversal potentials (and thus different driving forces). This becomes clearer when we inspect the ionic components of the synaptic current, which comprises an inward $Na^+$ current, an outward $K^+$

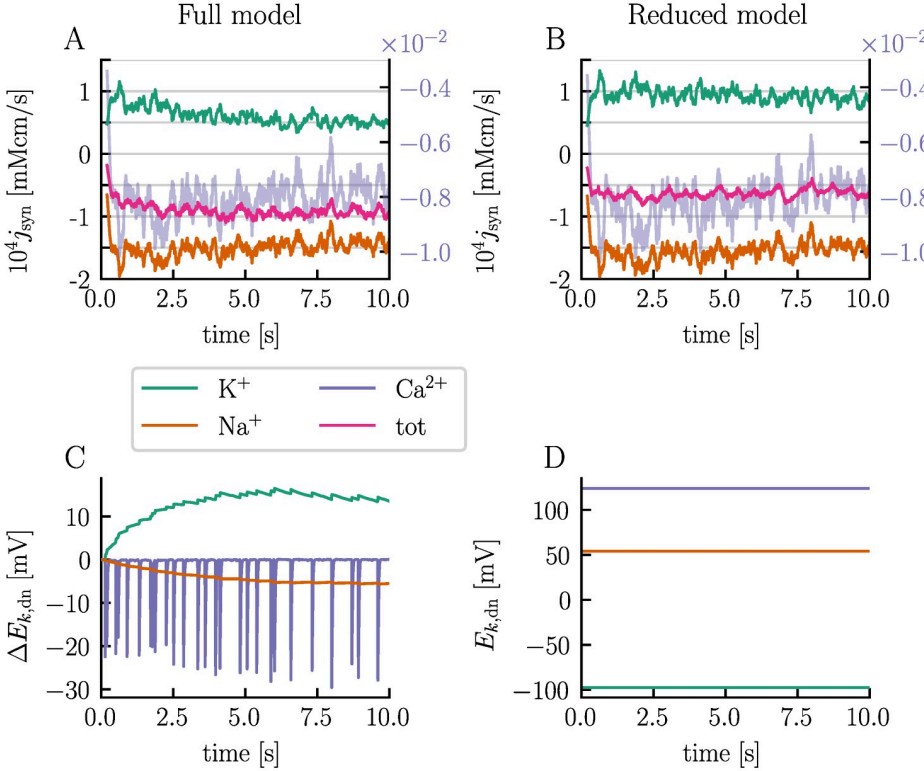

**Fig 6. Elevated ECS K⁺ concentrations strengthen the synapse.** The left panels illustrate the temporal sliding average of the synaptic flux divided into its ionic components (**A**) and the change in neuronal dendritic reversal potentials for K⁺, Na⁺, and Ca²⁺ (**C**) for the simulation presented in Fig 5 using the default modeling setup (full model). The right panels display the temporal sliding average of the synaptic flux divided into its ionic components (**B**) and the neuronal dendritic reversal potentials for K⁺, Na⁺, and Ca²⁺ (**D**) for the model ignoring ion concentration dynamics (reduced model). Sliding averages were calculated using a sliding window of 200 ms.

current, and a small inward Ca²⁺ current. In the full model, the K⁺ reversal potential gradually increases by approximately 15 mV, weakening the inhibitory effect of the K⁺ current (Fig 6A and 6C). The Na⁺ reversal potential slowly decreases by about 6 mV, slightly reducing the excitatory effect of the Na⁺ current (Fig 6A and 6C). The K⁺ current changes more than the Na⁺ current and the overall result is a strengthening of the synapse over time. In contrast, the ionic reversal potentials of the reduced model remain constant (Fig 6D). Consequently, the average synaptic current does not exhibit any notable strenghtening or weaking over time (Fig 6B).

## High ECS K⁺ concentrations induce spontaneous neuronal firing

In classical neuronal network models, synaptic connections determine how the neurons interact. In this paradigm, the ECS is assumed to be constant, and the spatial positioning of the neurons does not play a role. Instead, the crucial factor lies in identifying the specific connections between neurons. However, in real neuronal networks, communication may extend beyond synapses to include interactions through the ECS [44]. Previously, we saw how neurons in the electrodiffusive network model can interact via electric ephaptic coupling. In this context, the spatial configuration of the neurons becomes essential. Ephaptic coupling can also take ionic forms, meaning neurons can affect each other by altering the ionic environment of their neighbors [26, 44]. In this section, we will delve deeper into this aspect by conducting a case study.

Our study involves 25 units arranged in a linear configuration with a 1 μm spacing between them (Fig 7A). We eliminate all synaptic connections, so that any potential communication between neurons must occur solely via the ECS. To induce neuronal activity, we apply a strong, constant stimulus current to the neurons in units $1 - 15$, starting at $t = 0.1$ s.

Before stimulus onset, all neurons maintain a resting membrane potential of $-67$ mV. The stimulus current induces a rapid firing response in the neurons receiving stimuli, leading them into depolarization block approximately 5 seconds into the simulation, where they remain for the rest of the simulation (Fig 7B). The other neurons, belonging to units $16 - 25$, remain silent as long as neurons $1 - 15$ are active. However, we do detect signs of neighboring activity through subtle variations in their neuronal membrane potentials (Fig 7C). These variations are observed in all non-stimulated neurons, albeit with slight differences in amplitude. Over time, the non-stimulated neurons initially become more polarized, before their membrane potentials gradually increase, eventually surpassing the initial resting membrane potential (Fig 7B). The most noteworthy event occurs when neurons $16 - 25$ each fire an AP after about 9 seconds. Interestingly, this firing unfolds systematically: neuron 16, positioned closest to the cells in depolarization block, fires first. Subsequently, neurons 17, 18, and so forth follow in a sequential manner (Fig 7D).

The subtle variations observed in the membrane potentials of neurons $16 - 25$ during the firing of neurons $1 - 15$ result from electric ephaptic coupling. However, what ultimately triggers neurons $16 - 25$ to spontaneously fire while the system appears seemingly inactive is due to ionic effects, particularly changes in ECS $K^+$ concentrations. While neurons $1 - 15$ fire, the homeostatic machinery struggles to keep pace with the hyperactivity, leading to a continuous increase in ECS $K^+$ concentrations within the active region (Fig 8A). Given the interconnected nature of the ECS, ions migrate into the non-active region, distributing almost uniformly across units. By the time neuron $1 - 15$ enter depolarization block, the ECS $K^+$ concentration has inreased from 3.5 mM to approximately 8.5 mM across all units. In the depolarization block, the active $K^+$ channels of neurons $1 - 15$ remain in the open position, contributing to a continual rise in ECS $K^+$ concentrations. By the end of the simulation, the ECS $K^+$ concentration has increased to about 12.4 mM in all units. These concentration changes influence the neurons by altering the $K^+$ reversal potentials. Starting from a reversal potential of $-98$ mV, $E_K$ increases to about $-61$ mV and $-65$ mV by the end of the simulation for the stimulated and non-stimulated neurons, respectively (Fig 8B). The stimulated neurons experience the biggest change in $E_K$ because their intracellular $K^+$ concentrations change more due to their activity than those of the non-stimulated neurons. The rise in $E_K$ leads to an increase in the membrane potential of the neurons, eventually causing neurons $16 - 25$ to fire as the membrane potential crosses the firing threshold. It is worth noting that all other ion concentrations also change and influence the membrane potentials of the neurons. However, the ECS $K^+$ concentrations have the most substantial impact due to the large influence such changes have on $E_K$. Before moving on, we would like to emphasize that the ECS ion concentrations are not entirely uniform across the units. For units $16 - 25$, the ECS $K^+$ concentration is highest in unit 16, which is nearest to the active region, and lowest in unit 25, which is farthest from the active region (Fig 8C). This observation aligns with our expectations, considering that the increase is due to the migration of $K^+$ moving from the active to the non-active region. This subtle difference causes neuron 16 to fire an AP first, underscoring the influence of cell positioning.

## Astrocyte potassium buffering prevents neuronal depolarization blocks

Astrocytes are recognized for their role in maintaining a stable ionic environment for the neurons, particularly through a process termed potassium buffering [45–47]. In this process,

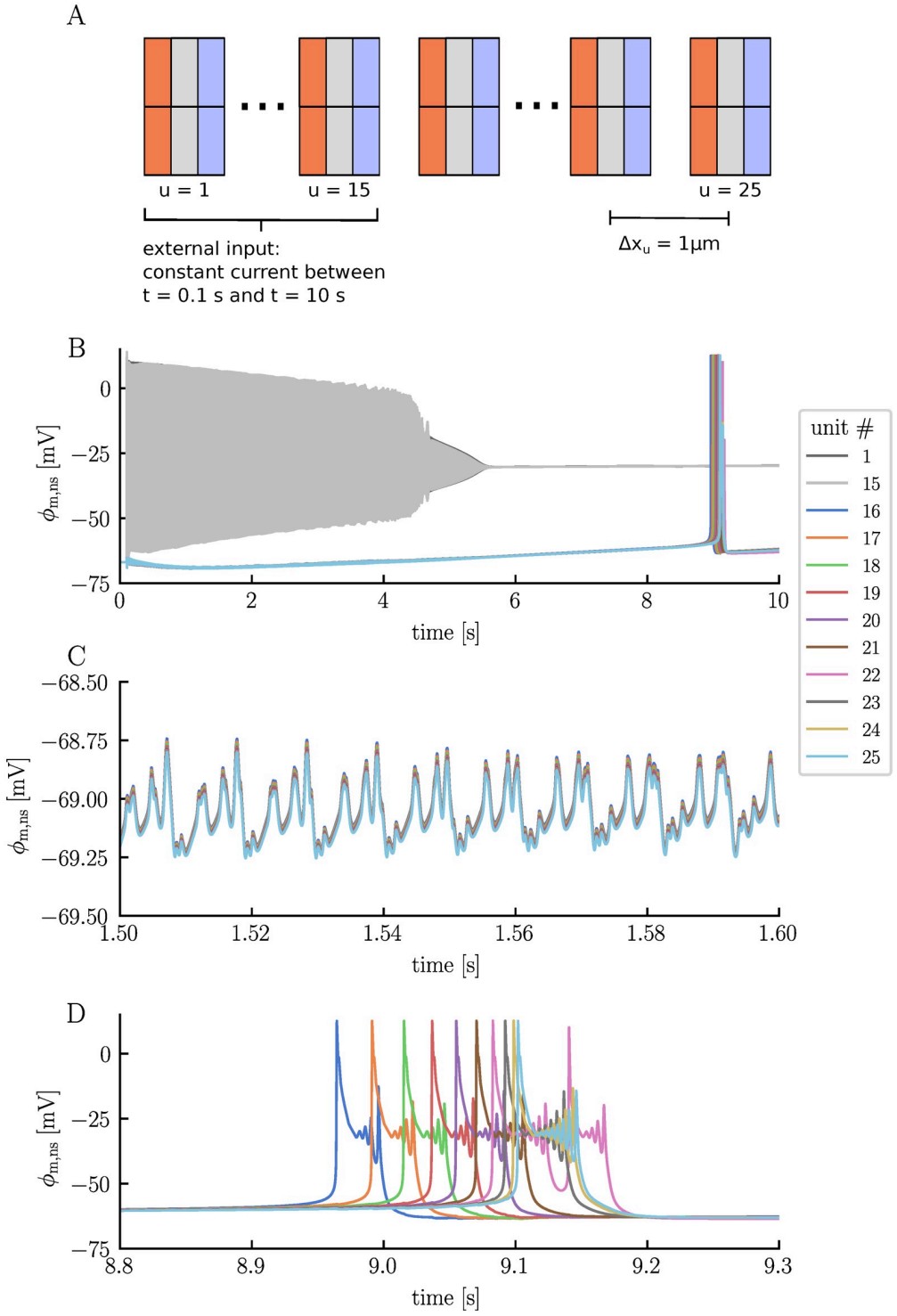

**Fig 7. The electrodiffusive network model in a linear configuration.** Panel **A** illustrates the simulation setup. 25 units are placed in a linear configuration, separated by a 1 μm distance. Neurons 1 − 15 are subject to a constant stimulus current, applied to the neuronal soma from $t = 0.1$ s to $t = 10$ s. Panel **B** illustrates the somatic membrane potential of neurons 1, 15, and 16 − 25. Neurons 2 − 14 are omitted from the panel but follow a firing pattern similar to neurons 1 and 15. Panels **C** and **D** display zoomed−in views of the membrane potentials of neurons 16 − 25.

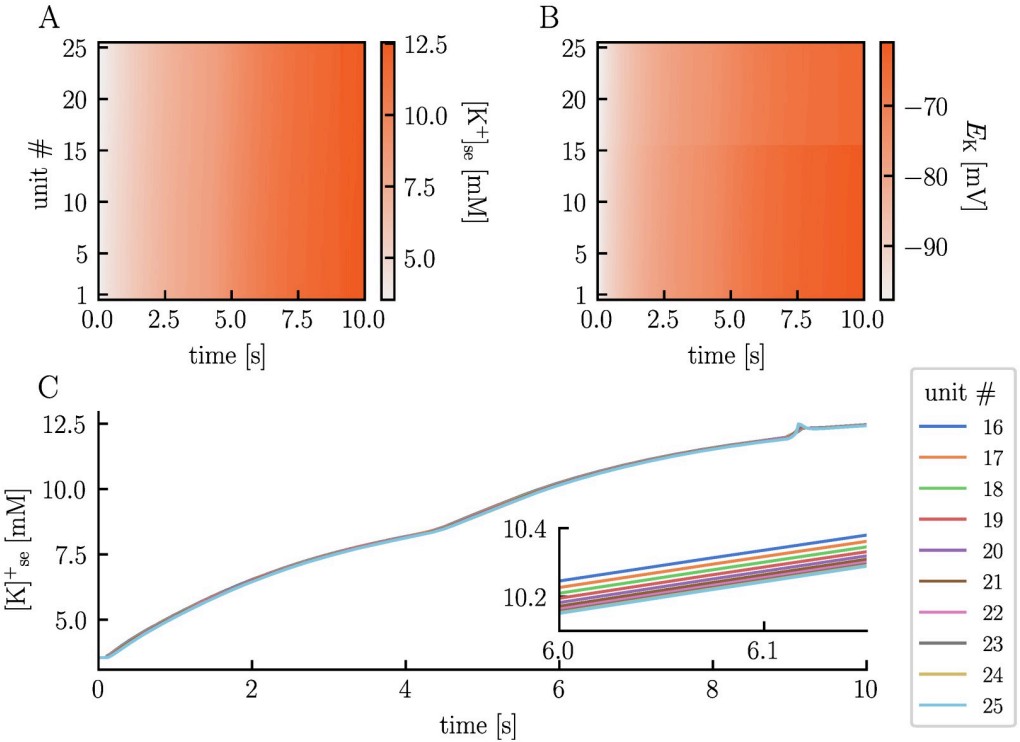

**Fig 8. Elevated ECS K$^+$ concentrations induce spontaneous neuronal firing.** The upper panels present heatmaps illustrating the ECS K$^+$ concentration in the soma layer (**A**) and the K$^+$ reversal potentials of the neuronal soma (**B**) across units (y-axes) and time (x-axes). Panel **C** displays the temporal evolution of the ECS K$^+$ concentration in the soma layer of units 16 − 25. The figure is based on the simulation presented in Fig 7.

astrocytes take up potassium from the ECS in areas where K$^+$ concentrations are high, transport it through the astrocyte syncytium, and release it in regions where ECS K$^+$ concentrations are low. To illustrate the role of glia and potassium buffering in the electrodiffusive network model, we run three simulations: one without glial membrane mechanisms (no-glia model), another with glial cells but weak gap junctions (weak-glia model), and a third one with glial cells and strong gap junctions (strong-glia model). In each simulation, we study two units separated by a 1 mm distance, with neuron 1 receiving a strong, constant stimulus current from $t = 0.1$ s to $t = 10$ s (Fig 9A). Conceptually, unit 1 can be viewed as being surrounded by numerous other neurons firing with an average frequency matching neuron 1. The second unit represents a region far away from the activity zone.

The different modeling setups lead to strikingly different firing patterns. In the weak-glia model, the stimulated neuron enters a depolarization block after approximately 3.5 seconds, where it remains for the rest of the simulation (Fig 9B). In the weak-glia model, the neuron similarly enters a depolarization block, but only after about 6 seconds (Fig 9C). Interestingly, in the strong-glia model, the neuron does not enter the depolarization block at all, but adapts to a lower, more sustainable firing rate after an initial period of hyperactivity (Fig 9D).

High ECS K$^+$ concentrations are known to trigger depolarization blocks [1, 48]. Indeed, the electrodiffusive network model predicts that efficient handling of ECS K$^+$ by the glial syncytium prevents such events. In the no-glia model, the ECS K$^+$ concentration in unit 1 exhibits a rapid increase (Fig 10B). As neuron 1 enters the depolarization block, the ECS K$^+$ concentration has increased by approximately 7.5 mM in the soma layer. The concentration continues

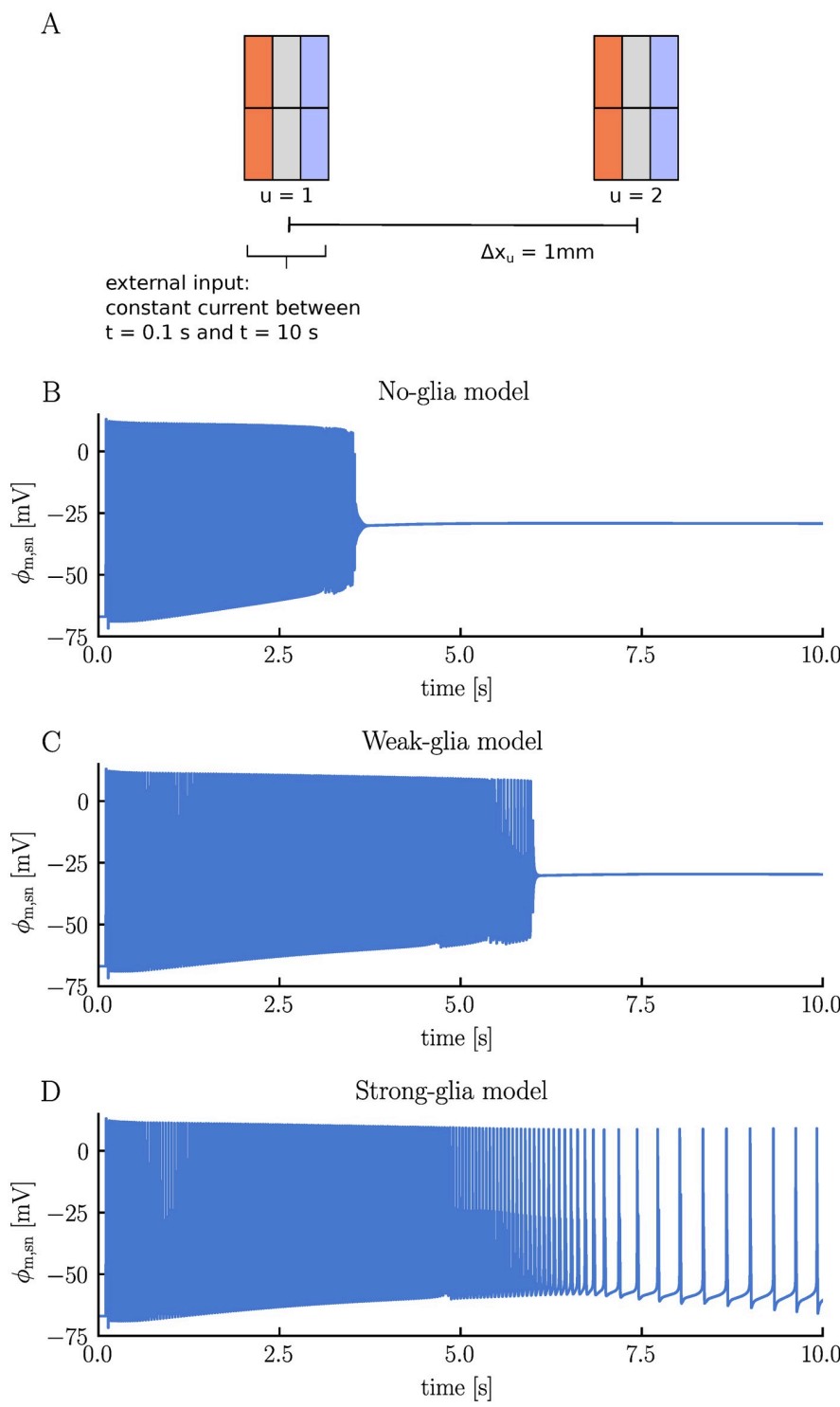

**Fig 9. Comparison of neuronal firing responses: no glia versus weak and strong glial coupling.** Panel **A** illustrates the simulation setup. Two units are placed next to each other, separated by a 1 mm distance. Neuron 1 receives a constant stimulus current between $t = 0.1$ s and $t = 10$ s. We run three simulations using different modeling setups: one with the glial membrane mechanisms turned off (no-glia model), another with glia present but featuring weak gap junctions (weak-glia model), and a third with glia present along with strong gap junctions (strong-glia model). All parameters are listed in Methods. Panels **B**, **C**, and **D** display the somatic membrane potential of neuron 1 for the no-glia, weak-glia, and strong-glia model, respectively.

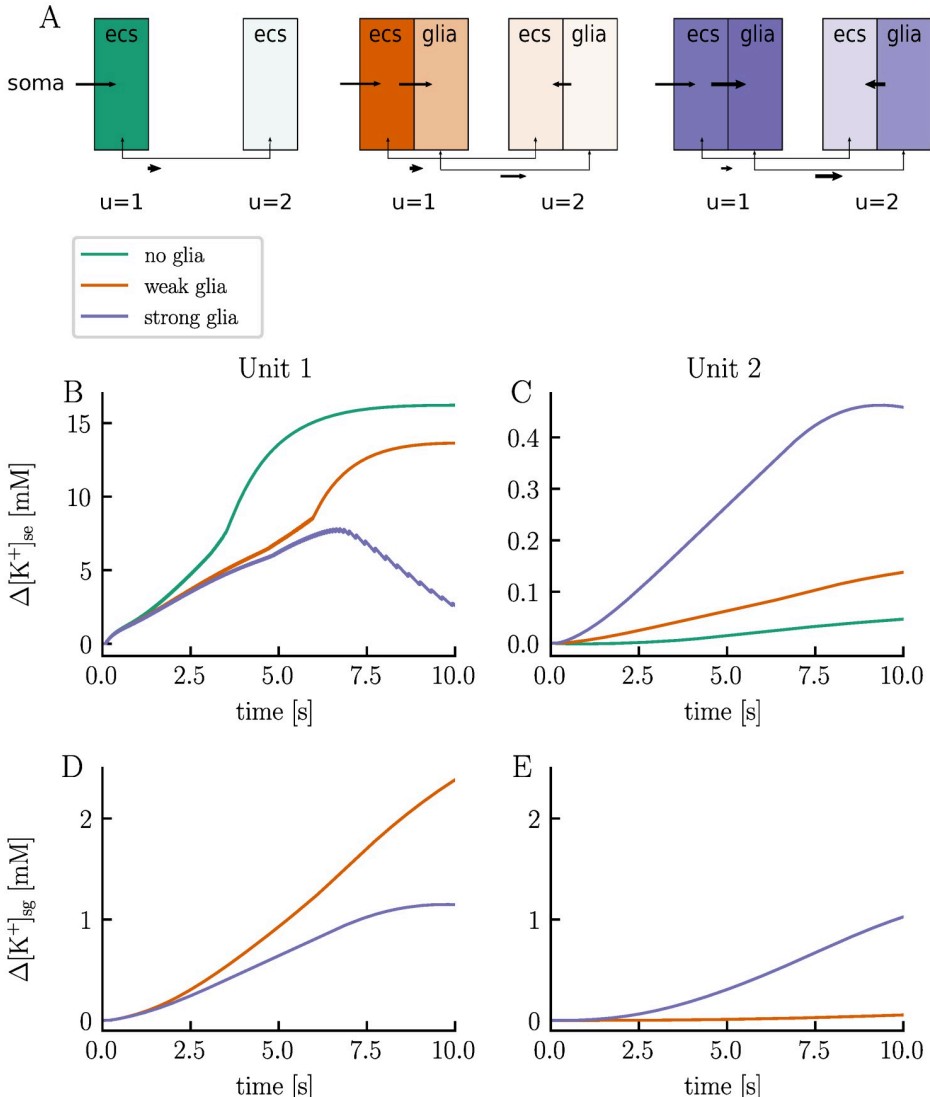

**Fig 10. Astrocyte potassium buffering prevents neuronal depolarization blocks.** Panel **A** shows a schematic overview of the distribution of excess $K^+$ ions in the ECS and glial cells in the soma layer at the simulation's endpoint for the no-glia (i.e., no glial membrane mechanisms), weak-glia (i.e., weak gap junctions), and strong-glia (i.e., strong gap junctions) model for the simulation presented in Fig 9. Stronger colors indicate a larger change from baseline values in terms of ion count. Additionally, arrows illustrates the direction of potassium transport, with longer arrows indicating more accumulated transport. When comparing two arrows of the same length, thicker arrows indicate greater transport. Panels **B**–**E** display the temporal evolution of the change in ECS $K^+$ concentrations in the soma layer of units 1 (**B**) and 2 (**C**), as well as the change in glial $K^+$ concentrations in the soma layer of units 1 (**D**) and 2 (**E**).

to climb, reaching a change of 16.2 mM by the end of the simulation. Given the absence of glia in this setup, excess $K^+$ can only be cleared through the ECS (aside from reuptake by the neuron and local electrodiffusion between layers). Due to the units' weak coupling (i.e., large $\Delta x_u$), extracellular transport is very slow. By the end of the simulation, the ECS $K^+$ concentration in the soma layer of unit 2 has increased by only 0.05 mM (Fig 10C). This transport is primarily diffusion driven (Fig 11A).

When we introduce weakly-coupled glia to the model, the ECS $K^+$ concentration in unit 1 initially increases at the same rate as in the no-glia model (Fig 10B). However, after about 0.5

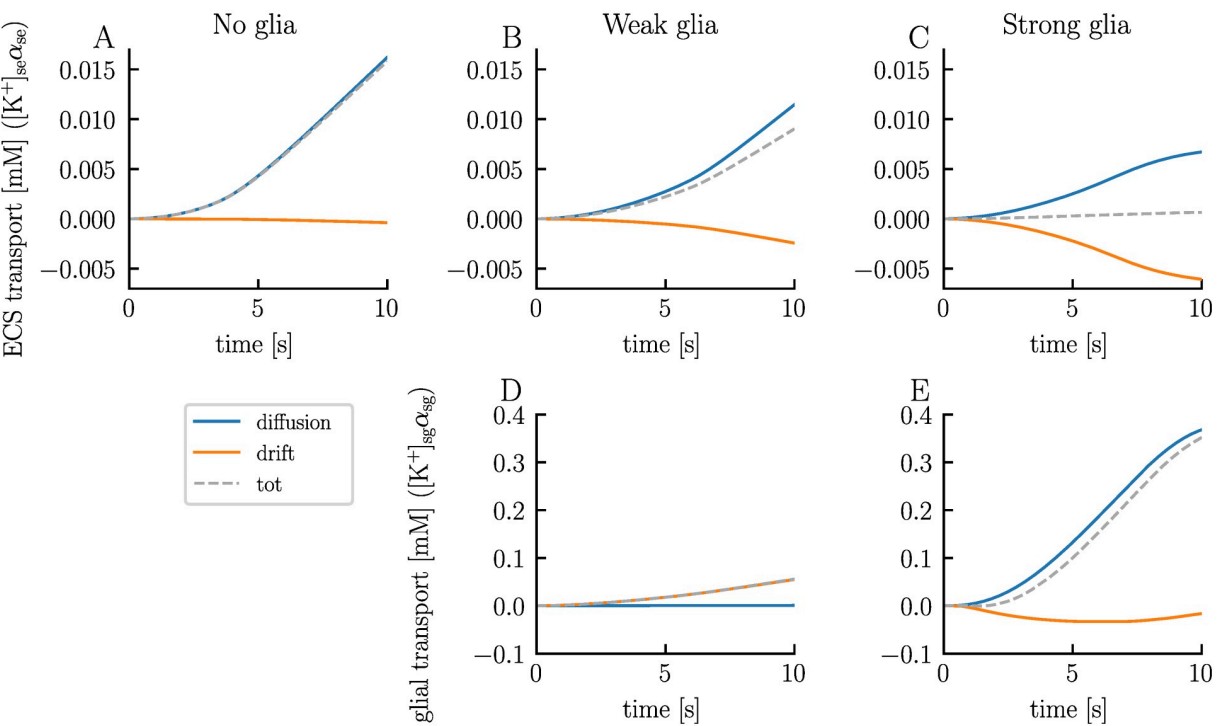

**Fig 11. Comparison of extracellular and glial K⁺ transport from unit 1 to unit 2: no glia versus weak and strong glial coupling.** The panels display the cumulative K⁺ transport from unit 1 to unit 2 over time (grey dashed lines), divided into its diffusive (blue lines) and electrical drift (orange lines) components, for the ECS in the no-glia model (**A**), weak-glia model (**B**), and strong-glia model (**C**), and for the glial domain in the weak-glia model (**D**) and strong-glia model (**E**). The figure is based on the simulation presented in Fig 9.

seconds, the ECS K⁺ concentration increases more gradually. By the end of the simulation, the concentration has increased by 13.6 mM in the soma layer—2.5 mM less than for the no-glia case. The ECS transport is on the same order of magnitude as in the no-glia model (Fig 11B). In fact, the absolute value is slightly lower, indicating that ECS transport is not the primary means of clearing K⁺. Instead, the slower increase in ECS K⁺ concentration in unit 1 results from glial uptake. Some of the absorbed K⁺ is subsequently transferred through the glial syncytium and released into the ECS of unit 2. By the end of the simulation, the K⁺ concentration in the soma layer has increased by 0.14 mM in the ECS of unit 2, 2.4 mM in the glial compartment of unit 1, and 0.005 mM in the glial compartment of unit 2 (Fig 10C, 10D and 10E, respectively). Note that the initial volume fraction of the glial compartments is double that of the extracellular compartments, meaning that a concentration increase of, say, 1 mM, in a glial compartment corresponds to an increase of 2 mM in an extracellular compartment in terms of ion count. In the weak-glia model, the extracellular transport is mainly diffusion driven, just like in the no-glia model (Fig 11B). Drift contributes slightly but works in the opposite direction of diffusion, slowing the transport. Interestingly, electric drift dominates the glial transport, being two orders of magnitude stronger than diffusion (Fig 11D). Here, both diffusion and electric drift work in the same direction.

In the strong-glia model, the ECS K⁺ concentration in the soma layer peaks with a change of 7.7 mM after 6.7 seconds, after which it starts to decrease (Fig 10A). At the end of the simulation, the concentration is only 2.6 mM above baseline value. Less K⁺ is transported through the ECS compared to the no-glia and weak-glia models, with diffusion and drift working in opposite directions at similar magnitudes (Fig 11C). The stronger glial coupling in this model

enhances the ability of the glia in unit 1 to absorb $K^+$, as excess intraglial $K^+$ can now be cleared more easily. As a result, glial uptake increases while less $K^+$ accumulates in the glia of unit 1 in the strong-glia model compared to the weak-glia model (Fig 10D). Additionally, more $K^+$ is released by the glia into the ECS of unit 2. By the end of the simulation, the $K^+$ concentration has increased by 0.46 mM in the ECS of unit 2, 1.2 mM in the glial compartment of unit 1, and 1.0 mM in the glial compartment of unit 2 (Fig 10C, 10D and 10E, respectively). In the strong-glia model, diffusion dominates the transport of $K^+$ through the glial network, being one order of magnitude stronger than the electric drift (Fig 11E). The electric drift works in the opposite direction of the diffusion, slowing the transport.

## Discussion

In this paper, we introduced the first electrodiffusive neuronal network model with multicompartmental neurons and synaptic connections. The model comprises an arbitrary number of "units," each representing a neuron and its immediate surroundings, that is, a piece of glia and extracellular space. The units are arranged in a linear (i.e., 1D) configuration. Unlike conventional network models that focus primarily on neuronal spiking patterns, the electrodiffusive network model predicts ion concentrations ($Na^+$, $K^+$, $Cl^-$, and $Ca^{2+}$), electrical potentials, and volume fractions in all compartments, both intra- and extracellularly. While developing the model, our focus has been on biophysical rigor and mechanistic understanding, rather than on morphological detail or excessive complexity. This approach has resulted in a model that is relatively simple, yet it predicts ion concentrations, electrical potentials, and volume changes in a manner that aligns with first principles. We acknowledge that the choice to position the units in a 1D configuration can lead to somewhat high amplitudes in the extracellular AP signatures and exaggerated ion concentration changes. This outcome is a consequence of the increased total resistance (limited degrees of freedom) for extracellular currents and a reduction in the volume of tissue available. Even though the volumes did not change dramatically in the test cases explored in this paper, we decided to incorporate volume dynamics into the model. This inclusion was due to the potential for volumes to notably change and influence the ion concentration dynamics in other scenarios. Indeed, we showed in a previous study that models without volume dynamics severely overestimate osmotic pressures as compared to models that include volume dynamics [33].

The unique properties of the electrodiffusive network model lead to behaviors not captured by conventional models based on cable theory, where ion concentrations are assumed constant, and the extracellular potential is set to zero. First, we observed that changes in the ion concentrations affected the synaptic strengths. In a case study where we examined a single neuron, with and without ion concentration dynamics, receiving a 400 Hz Poisson spike train, the model with ion concentration dynamics exhibited a higher firing rate than the model without ion concentration dynamics. Upon closer examination, we observed that in the model with ion concentration dynamics, the synaptic current increased over time due to shifts in the ionic reversal potentials.

In another case study involving 25 units without synaptic connections, where the first 15 neurons received a strong, constant stimulus current that eventually led the neurons into depolarization block, we observed how ephaptic effects—both electric and ionic—influenced the system. The electric ephaptic effects were evident in the model each time a neuron fired an action potential. When this happened, it resulted in a subtle change in the extracellular potential across all units, with the amplitude of the change being dependent on the distance to the firing neuron. This shift, in turn, influenced the membrane potential of all other neurons and the glial syncytium. The ionic ephaptic effects were slower, but more dramatic. The firing

neurons gradually affected the ionic environment of all the other cells in the system, thereby influencing their ionic reversal- and membrane potentials. This eventually resulted in spontaneous firing of those neurons that were not receiving any stimuli, despite the absence of synaptic connections.

In a final test case, we took a closer look at the effects of incorporating glia into the network. By comparing a model without glia to models with weakly- and strongly coupled glial syncytium, we observed that efficient potassium buffering can prevent neuronal depolarization blocks.

We envision that the electrodiffusive network model will benefit the neuroscience community in three main ways. Firstly, by relaxing some common model assumptions, the model can be used to evaluate the validity of these assumptions under various conditions. For instance, we can determine when it is necessary to consider the effect of electric drift on ion concentrations or when to take into account the impact of diffusive currents on extracellular potentials. This is a topic we have also discussed in previous work for single cells [1, 28]. Secondly, we think that our model is foundational in its incorporation of all relevant physics and can thus lead to a deeper understanding and intuition of electrophysiology. Thirdly, the electrodiffusive network model can be useful for exploring a variety of scenarios influenced by ion concentration dynamics. Any desired changes to the membrane mechanisms or parameters can be readily made while maintaining the model's core framework. Additionally, while we have utilized excitatory synapses in this paper, the model can easily accommodate inhibitory synapses or a combination of both. The only requirement is that the synapse models ensure ion conservation. One could also enhance the synapse models by incorporating elements such as GABA- or glutamate-dependency, taking cues from models like the one presented in Tuttle et al. 2019 [32]. An intriguing extension of the model could involve expanding its dimensionality from one to two or three dimensions, allowing for a more realistic spatial representation of the network. This could be achieved by positioning the units in a 2D or 3D grid structure. Expanding the model's dimensionality would not present a substantial mathematical challenge; however, large-scale network simulations would necessitate careful consideration during model implementation to keep computational demands within manageable limits. This is particularly crucial when calculating the synaptic conductances [49].

## Methods

The electrodiffusive network model comprises an arbitrary number ($U$) of "units" (denoted by superscript $u$), each representing a neuron and its immediate surroundings (Fig 1). The units can be arranged either linearly (i.e., with closed boundary conditions) or in a ring (i.e., with periodic boundary conditions). Each unit is divided into two layers: a soma layer (denoted by subscript $l = s$), and a dendrite layer (denoted by subscript $l = d$). These layers are further subdivided into three domains, resulting in a total of six compartments in each unit. Specifically, two compartments are dedicated to representing the neuron (denoted by subscript $r = n$), another two represent glial cells (denoted by subscript $r = g$), and the remaining two represent ECS (denoted by subscript $r = e$). The model predicts the temporal evolution of the volume fractions $\alpha_{rl}^u$, ion concentrations $[\text{Na}^+]_{rl}^u$, $[\text{K}^+]_{rl}^u$, $[\text{Cl}^-]_{rl}^u$, and $[\text{Ca}^{2+}]_{rl}^u$, and electrical potentials $\phi_{rl}^u$ and is implemented using the KNP framework, ensuring a biophysically consistent relationships between ion concentrations, electrical potentials, and charge [29, 41, 50]. A list of symbols and definitions are given in Table 1 and model parameters are listed in Table 2.

**Table 1. List of symbols and definitions.**

| Symbol | Definiton | Units |
|---|---|---|
| $\tau$ (index) | Time step: 1, 2, . . ., $T$ | – |
| $u$ (index) | Unit number: 1, 2, . . ., $U$ | – |
| $l$ (index) | Layer: s (soma) or d (dendrite) | – |
| $r$ (index) | Domain: n (neuron), e (ECS), or g (glia) | – |
| $k$ (index) | Ion species: Na$^+$, K$^+$, Cl$^-$, or Ca$^{2+}$ | – |
| $\alpha_{rl}^{u,\tau}$ | Volume fraction | – |
| $[k]_{rl}^{u,\tau}$ | Ion concentration | mM |
| $\phi_{rl}^{u,\tau}$ | Electrical potential | mV |
| $\phi_{\mathrm{m},rl}^{u,\tau}$ | Membrane potential | mV |
| $w_{\mathrm{m},rl}^{u,\tau}$ | Transmembrane water velocity | cm/s |
| $j_{\mathrm{m},k,rl}^{u,\tau}$ | Membrane flux density | mMcm/s |
| $j_{\mathrm{interunits},k,rl}^{u,\tau}$ | Ion flux density between units | mMcm/s |
| $j_{\mathrm{interlayers},k,r}^{u,\tau}$ | Ion flux density between layers | mMcm/s |
| $a_{rl}^{u}$ | Amount of immobile ions | mM |
| $E_{k,rl}^{u,\tau}$ | Reversal potential | mV |

**Table 2. Model parameters.** All model parameters are taken from Sætra et al. 2021 [28].

| Symbol | Definition | Value |
|---|---|---|
| $\gamma_{\mathrm{m}}$ | Membrane area per unit volume of tissue | $1.714683 \cdot 10^3$ 1/cm |
| $\Delta x_{\mathrm{l}}$ | Distance between layers | $6.67 \cdot 10^{-2}$ cm |
| $D_{\mathrm{Na}}$ | Na$^+$ diffusion coefficient | $1.33 \cdot 10^{-5}$ cm$^2$/s |
| $D_{\mathrm{K}}$ | K$^+$ diffusion coefficient | $1.96 \cdot 10^{-5}$ cm$^2$/s |
| $D_{\mathrm{Cl}}$ | Cl$^-$ diffusion coefficient | $2.03 \cdot 10^{-5}$ cm$^2$/s |
| $D_{\mathrm{Ca}}$ | Ca$^{2+}$ diffusion coefficient | $0.71 \cdot 10^{-5}$ cm$^2$/s |
| $v_{\{\mathrm{Na,K,Cl}\},\{\mathrm{n,e,g}\}}$ | Na$^+$, K$^+$, and Cl$^-$ mobility fractions | 1 |
| $v_{\mathrm{Ca,n}}$ | Ca$^{2+}$ mobility fraction (n) | 0.01 |
| $v_{\mathrm{Ca,e}}$ | Ca$^{2+}$ mobility fraction (e) | 1 |
| $\lambda_{\{\mathrm{n,g}\}}$ | Intracellular tortuosity | 3.2 |
| $\lambda_{\mathrm{e}}$ | Extracellular tortuosity | 1.6 |
| $\theta_r$ | Coupling strength between layers | 350$^\dagger$ |
| $F$ | Faraday constant | 96.48 C/mmol |
| $R$ | Gas constant | 8.314 J/(molK) |
| $T$ | Temperature | 309.14 K |
| $C_{\mathrm{m}}$ | Membrane capacitance | $3 \cdot 10^{-6}$ F/cm$^2$ |
| $\eta_{\mathrm{m,n}}$ | membrane water permeability (n) | $3.25 \cdot 10^{-12}$ cm/(Pas) |
| $\eta_{\mathrm{m,g}}$ | membrane water permeability (g) | $8.12 \cdot 10^{-12}$ cm/(Pas) |
| $i$ | Van't Hoff factor | 1 |
| $\bar{g}_{\mathrm{Na}}$ | max Na$^+$ conductance | $3 \cdot 10^{-2}$ S/cm$^2$ |
| $\bar{g}_{\mathrm{DR}}$ | max DR conductance | $1.5 \cdot 10^{-2}$ S/cm$^2$ |
| $\bar{g}_{\mathrm{Ca}}$ | max Ca$^{2+}$ conductance | $1.18 \cdot 10^{-2}$ S/cm$^2$ |
| $\bar{g}_{\mathrm{AHP}}$ | max AHP conductance | $8 \cdot 10^{-4}$ S/cm$^2$ |
| $\bar{g}_{\mathrm{C}}$ | max C conductance | $1.5 \cdot 10^{-2}$ S/cm$^2$ |
| $\bar{g}_{\mathrm{leak,Na,n}}$ | Na$^+$ leak conductance (n) | $2.46 \cdot 10^{-5}$ S/cm$^2$ |
| $\bar{g}_{\mathrm{leak,K,n}}$ | K$^+$ leak conductance (n) | $2.45 \cdot 10^{-5}$ S/cm$^2$ |

*(Continued)*

**Table 2.** (Continued)

| Symbol | Definition | Value |
|---|---|---|
| $\bar{g}_{\text{leak,Cl,n}}$ | Cl$^-$ leak conductance (n) | $1 \cdot 10^{-4}$ S/cm$^2$ |
| $\rho_{\text{n}}$ | pump strength (n) | $1.87 \cdot 10^{-4}$ mMcm/s |
| $U_{\text{kcc2}}$ | kcc2 cotransporter strength | $1.49 \cdot 10^{-5}$ mMcm/s |
| $U_{\text{nkcc1}}$ | nkcc1 cotransporter strength | $2.33 \cdot 10^{-5}$ mMcm/s |
| $U_{\text{Ca-dec}}$ | Ca$^{2+}$ decay rate | $75$ s$^{-1}$ |
| $[\text{Ca}^{2+}]^u_{\text{basal},nl}$ | basal Ca$^{2+}$ concentration (n) | $0.01$ mM |
| $\bar{g}_{\text{leak,Na,g}}$ | Na$^+$ leak conductance (g) | $1 \cdot 10^{-4}$ S/cm$^2$ |
| $\bar{g}_{\text{leak,Cl,g}}$ | Cl$^-$ leak conductance (g) | $5 \cdot 10^{-5}$ S/cm$^2$ |
| $\bar{g}_{\text{Kir}}$ | max Kir conductance | $1.696 \cdot 10^{-3}$ S/cm$^2$ |
| $[\text{K}^+]^u_{\text{basal},el}$ | basal K$^+$ concentration (e) | $3.082$ mM |
| $[\text{K}^+]^u_{\text{basal},gl}$ | basal K$^+$ concentration (g) | $99.959$ mM |
| $\rho_{\text{g}}$ | pump strength (g) | $1.12 \cdot 10^{-4}$ mMcm/s |
| $[\text{Na}^+]_{\text{threshold,g}}$ | glial pump Na$^+$ threshold | $10$ mM |
| $[\text{K}^+]_{\text{threshold,g}}$ | glial pump K$^+$ threshold | $1.5$ mM |
| $\bar{g}_{\text{syn,Na}}$ | max synaptic conductance (Na$^+$) | $1.623 \cdot 10^{-4}$ S/cm$^2$ |
| $\bar{g}_{\text{syn,K}}$ | max synaptic conductance (K$^+$) | $3.084 \cdot 10^{-4}$ S/cm$^2$ |
| $\bar{g}_{\text{syn,Ca}}$ | max synaptic conductance (Cl$^-$) | $1.055 \cdot 10^{-6}$ S/cm$^2$ |
| $\tau_1$ | synaptic decay time constant | $0.003$ s |
| $\tau_2$ | synaptic rise time constant | $0.001$ s |

†Manually tuned to achieve an AP shape similar to the edNEG model [28].

## Governing equations

We model the network using a system of discretized, coupled, nonlinear ODEs. At time $t = \tau\Delta t$, the intracellular volume fractions $\alpha_{rl}^{u,\tau}$ (unitless), where $r = \{\text{n, g}\}$, $l = \{\text{s, d}\}$, and $u = \{1, \ldots, U\}$, are given by:

$$\frac{\alpha_{rl}^{u,\tau} - \alpha_{rl}^{u,\tau-1}}{\Delta t} = -\gamma_{\text{m}} w_{\text{m},rl}^{u,\tau}. \tag{1}$$

Here, the coefficient $\gamma_{\text{m}}$ (1/cm) represents the cell membrane area per unit volume of tissue and $w_{\text{m},rl}^{u,\tau}$ (cm/s) is the transmembrane water velocity. We assume $\gamma_{\text{m}}$ to be constant in time and the same for all compartments. By definition, the volume fractions in a specific layer within a given unit always sum to 1. We thus have that the extracellular volume fractions $\alpha_{el}^{u,\tau}$ (unitless), where $l = \{\text{s, d}\}$ and $u = \{1, \ldots, U\}$, are given by:

$$\alpha_{el}^{u,\tau} = 1.0 - (\alpha_{nl}^{u,\tau} + \alpha_{gl}^{u,\tau}). \tag{2}$$

Further, for each ion species $k \in \{\text{Na}^+, \text{K}^+, \text{Cl}^-, \text{Ca}^{2+}\}$, we find the ion concentrations $[k]_{rl}^{u,\tau}$ (mM), where $r = \{\text{n, g, e}\}$, $l = \{\text{s, d}\}$, and $u = \{1, \ldots, U\}$, by solving the following equations:

$$\frac{\alpha_{\text{ns}}^{u,\tau}[k]_{\text{ns}}^{u,\tau} - \alpha_{\text{ns}}^{u,\tau-1}[k]_{\text{ns}}^{u,\tau-1}}{\Delta t} = -\frac{\alpha_{\text{ns}}^{u,\tau-1}}{\Delta x_{\text{l}}} j_{\text{interlayers},k,\text{n}}^{u,\tau} - \gamma_{\text{m}} j_{\text{m},k,\text{ns}}^{u,\tau}, \tag{3a}$$

$$\frac{\alpha_{\text{nd}}^{u,\tau}[k]_{\text{nd}}^{u,\tau} - \alpha_{\text{nd}}^{u,\tau-1}[k]_{\text{nd}}^{u,\tau-1}}{\Delta t} = \frac{\alpha_{\text{ns}}^{u,\tau-1}}{\Delta x_{\text{l}}} j_{\text{interlayers},k,\text{n}}^{u,\tau} - \gamma_{\text{m}} j_{\text{m},k,\text{nd}}^{u,\tau}, \tag{3b}$$

$$\frac{\alpha_{es}^{u,\tau}[k]_{es}^{u,\tau} - \alpha_{es}^{u,\tau-1}[k]_{es}^{u,\tau-1}}{\Delta t} = \frac{\alpha_{es}^{u-1,\tau-1}}{\Delta x_u} j_{\text{interunits},k,es}^{u-1,\tau} - \frac{\alpha_{es}^{u,\tau-1}}{\Delta x_u} j_{\text{interunits},k,es}^{u,\tau}$$
$$- \frac{\alpha_{es}^{u,\tau-1}}{\Delta x_l} j_{\text{interlayers},k,e}^{u,\tau} + \gamma_m \left( j_{m,k,ns}^{u,\tau} + j_{m,k,gs}^{u,\tau} \right), \tag{3c}$$

$$\frac{\alpha_{ed}^{u,\tau}[k]_{ed}^{u,\tau} - \alpha_{ed}^{u,\tau-1}[k]_{ed}^{u,\tau-1}}{\Delta t} = \frac{\alpha_{ed}^{u-1,\tau-1}}{\Delta x_u} j_{\text{interunits},k,ed}^{u-1,\tau} - \frac{\alpha_{ed}^{u,\tau-1}}{\Delta x_u} j_{\text{interunits},k,ed}^{u,\tau}$$
$$+ \frac{\alpha_{es}^{u,\tau-1}}{\Delta x_l} j_{\text{interlayers},k,e}^{u,\tau} + \gamma_m \left( j_{m,k,nd}^{u,\tau} + j_{m,k,gd}^{u,\tau} \right), \tag{3d}$$

$$\frac{\alpha_{gs}^{u,\tau}[k]_{gs}^{u,\tau} - \alpha_{gs}^{u,\tau-1}[k]_{gs}^{u,\tau-1}}{\Delta t} = \frac{\alpha_{gs}^{u-1,\tau-1}}{\Delta x_u} j_{\text{interunits},k,gs}^{u-1,\tau} - \frac{\alpha_{gs}^{u,\tau-1}}{\Delta x_u} j_{\text{interunits},k,gs}^{u,\tau}$$
$$- \frac{\alpha_{gs}^{u,\tau-1}}{\Delta x_l} j_{\text{interlayers},k,g}^{u,\tau} - \gamma_m j_{m,k,gs}^{u,\tau}, \tag{3e}$$

$$\frac{\alpha_{gd}^{u,\tau}[k]_{gd}^{u,\tau} - \alpha_{gd}^{u,\tau-1}[k]_{gd}^{u,\tau-1}}{\Delta t} = \frac{\alpha_{gd}^{u-1,\tau-1}}{\Delta x_u} j_{\text{interunits},k,gd}^{u-1,\tau} - \frac{\alpha_{gd}^{u,\tau-1}}{\Delta x_u} j_{\text{interunits},k,gd}^{u,\tau}$$
$$+ \frac{\alpha_{gs}^{u,\tau-1}}{\Delta x_l} j_{\text{interlayers},k,g}^{u,\tau} - \gamma_m j_{m,k,gd}^{u,\tau}. \tag{3f}$$

Here, $\Delta x_l$ (cm) is the distance between the layers and $\Delta x_u$ (cm) is the distance between units. The equations say that the ion concentration in a given compartment is determined by the balance of ionic fluxes entering and leaving the compartment, ensuring ion conservation. The ionic fluxes constitute three types: transmembrane fluxes $j_{m,k,rl}^{u,\tau}$ (mMcm/s), fluxes between units $j_{\text{interunits},k,rl}^{u,\tau}$ (mMcm/s), and fluxes between layers $j_{\text{interlayers},k,r}^{u,\tau}$ (mMcm/s). We define positive directions to go from intracellular to extracellular, from soma to dendrite, and from unit $u$ to unit $u + 1$.

The ionic fluxes between units and layers are given by the Nernst-Planck equation, which accounts for the movement of ions due to both diffusion and electric drift:

$$j_{\text{interunits},k,rl}^{u,\tau} = -\frac{D_k v_{k,r}}{\lambda_r^2} \left( \frac{[k]_{rl}^{u+1,\tau} - [k]_{rl}^{u,\tau}}{\Delta x_u} \right)$$
$$- \frac{D_k z_k v_{k,r}}{\lambda_r^2 \psi} \left( \frac{[k]_{rl}^{u,\tau} + [k]_{rl}^{u+1,\tau}}{2} \right) \left( \frac{\phi_{rl}^{u+1,\tau} - \phi_{rl}^{u,\tau}}{\Delta x_u} \right), \tag{4a}$$

$$j_{\text{interlayers},k,r}^{u,\tau} = -\frac{\theta_r D_k v_{k,r}}{\lambda_r^2} \left( \frac{[k]_{rd}^{u,\tau} - [k]_{rs}^{u,\tau}}{\Delta x_l} \right)$$
$$- \frac{\theta_r D_k z_k v_{k,r}}{\lambda_r^2 \psi} \left( \frac{[k]_{rs}^{u,\tau} + [k]_{rd}^{u,\tau}}{2} \right) \left( \frac{\phi_{rd}^{u,\tau} - \phi_{rs}^{u,\tau}}{\Delta x_l} \right). \tag{4b}$$

Here, $D_k$ (cm²/s) is the diffusion coefficient of ion species $k$, $v_{k,r}$ (unitless) is the fraction of free ions (= 1 for all ions, except intraneuronal $Ca^{2+}$ which is partly buffered or taken up by the endoplasmic reticulum), $\lambda_r$ (unitless) is the tortuosity of domain $r$, $z_k$ (unitless) is the valency of ion species $k$, $\theta_r$ (unitless) is the coupling strength between the layers, and $\psi = RT/F$ (mV), where $R$ (J/(molK)) is the gas constant, $T$ (K) is the absolute temperature, and $F$ (C/mmol) is Faraday's constant. For the neurons, ions can move intracellularly between the soma and

dendrite layers but not directly between different units without passing through the extracellular space (i.e, $j_{\text{interunits},k,\text{nl}}^{u,\tau} = 0$). In the ECS, ions are free to move both between layers and different units. We assume that the glial domain represents an astrocyte syncytium connected through gap junctions, which permits ions to move in the same manner as in the ECS.

To calculate the electrical potentials $\phi_{rl}^{u,\tau}$ (mV), where $r = \{\text{n, g, e}\}$, $l = \{\text{s, d}\}$ and $u = \{1, \ldots, U\}$, we use the charge-capacitor relation, assuming electroneutrality of the bulk solution:

$$\gamma_{\text{m}} C_{\text{m}} \phi_{\text{m},rl}^{u,\tau} = z_0 F a_{rl}^u + \alpha_{rl}^{u,\tau} F \sum_k z_k [k]_{rl}^{u,\tau}, \quad r = \text{n, g} \tag{5a}$$

$$-\gamma_{\text{m}} C_{\text{m}} (\phi_{\text{m},nl}^{u,\tau} + \phi_{\text{m},gl}^{u,\tau}) = z_0 F a_{el}^u + \alpha_{el}^{u,\tau} F \sum_k z_k [k]_{el}^{u,\tau}, \tag{5b}$$

where, $\phi_{\text{m},rl}^{u,\tau} = \phi_{rl}^{u,\tau} - \phi_{el}^{u,\tau}$. The parameter $C_{\text{m}}$ (F/cm$^2$) is the membrane capacitance, $a_{rl}^u$ (mM) denotes the concentration of immobile ions, and $z_0$ (unitless) is the valency of $a$. The electrical potential is only determined up to a constant. For this reason, we require that:

$$\phi_{ed}^{0,\tau} = 0. \tag{6}$$

The combination of Eqs 1, 3 and 5, with insertion of Eq 4, defines a system of $U \times 34$ equations for the $U \times 34$ unknowns $\alpha_{\{n,g\}l}^{u,\tau}$, $[k]_{rl}^{u,\tau}$, and $\phi_{rl}^{u,\tau}$. In addition, we need to solve a set of $U \times 5$ ODEs to describe the membrane fluxes (c.f. the Neuronal membrane mechanisms section). We find the extracellular volumes using Eq 2. Appropriate initial conditions and boundary conditions close the system.

## Transmembrane fluid velocity

The transmembrane fluid flow is driven by osmotic pressure gradients and a constant transmembrane fluid pressure, which we set to ensure zero fluid flow at $t = 0$ s. The fluid velocity $w_{\text{m},rl}^{u,\tau}$ (m/s) is then given by:

$$w_{\text{m},rl}^{u,\tau} = \eta_{\text{m},r} \left( \Delta p_{rl}^u + iRT \left( \frac{a_{el}^u}{\alpha_{el}^{u,\tau}} + \sum_k \upsilon_{k,r} [k]_{el}^{u,\tau} - \frac{a_{rl}^u}{\alpha_{rl}^{u,\tau}} - \sum_k \upsilon_{k,r} [k]_{rl}^{u,\tau} \right) \right), \tag{7}$$

where $\eta_{\text{m},r}$ (cm/(Pas)) is the membrane water permeability, $\Delta p_{rl}^u$ (Pa) is the transmembrane fluid pressure, and $i$ (unitless) is the Van't Hoff factor. The transmembrane fluid pressure is expressed as:

$$\Delta p_{rl}^u = -iRT \left( \frac{a_{el}^u}{\alpha_{el}^{u,0}} + \sum_k \upsilon_{k,r} [k]_{el}^{u,0} - \frac{a_{rl}^u}{\alpha_{rl}^{u,0}} - \sum_k \upsilon_{k,r} [k]_{rl}^{u,0} \right). \tag{8}$$

## Neuronal membrane mechanisms

We adopt the neuronal membrane mechanisms from the edNEG model [28]. The active ion channels include a voltage-dependent Na$^+$ channel (Na) and a voltage-dependent K$^+$ delayed rectifier channel (DR) in the soma, and a voltage-dependent Ca$^{2+}$ channel (Ca), a K$^+$ afterhyperpolarization channel (AHP), and a Ca$^{2+}$-dependent K$^+$ channel (C) in the dendrite. The

ionic fluxes (mMcm/s) are given by:

$$j_{\text{Na}}^{u,\tau} = g_{\text{Na}}^{\tau}(\phi_{\text{m,ns}}^{u,\tau} - E_{\text{Na,ns}}^{u,\tau})/(Fz_{\text{Na}}),\tag{9a}$$

$$j_{\text{DR}}^{u,\tau} = g_{\text{DR}}^{\tau}(\phi_{\text{m,ns}}^{u,\tau} - E_{\text{K,ns}}^{u,\tau})/(Fz_{\text{K}}),\tag{9b}$$

$$j_{\text{Ca}}^{u,\tau} = g_{\text{Ca}}^{\tau}(\phi_{\text{m,nd}}^{u,\tau} - E_{\text{Ca,nd}}^{u,\tau})/(Fz_{\text{Ca}}),\tag{9c}$$

$$j_{\text{AHP}}^{u,\tau} = g_{\text{AHP}}^{\tau}(\phi_{\text{m,nd}}^{u,\tau} - E_{\text{K,nd}}^{u,\tau})/(Fz_{\text{K}}),\tag{9d}$$

$$j_{\text{C}}^{u,\tau} = g_{\text{C}}^{\tau}(\phi_{\text{m,nd}}^{u,\tau} - E_{\text{K,nd}}^{u,\tau})/(Fz_{\text{K}}).\tag{9e}$$

In Eq 9, $g_{\text{Na}}^{\tau}$, $g_{\text{DR}}^{\tau}$, $g_{\text{Ca}}^{\tau}$, $g_{\text{AHP}}^{\tau}$, and $g_{\text{C}}^{\tau}$ (S/cm$^2$) are the ion conductances, $\phi_{\text{m},nl}^{u,\tau}$ (mV) is the neuronal membrane potential defined as $\phi_{nl}^{u,\tau} - \phi_{el}^{u,\tau}$, and $E_{k,nl}^{u,\tau}$ (mV) is the reversal potential of ion species $k$. The reversal potentials are given by the Nernst equation:

$$E_{k,rl}^{u,\tau} = \frac{RT}{z_k F}\ln\frac{[k]_{el}^{u,\tau}}{v_{k,r}[k]_{rl}^{u,\tau}}.\tag{10}$$

For the active conductances, we have that:

$$g_{\text{Na}}^{\tau} = \bar{g}_{\text{Na}}(m_{\infty}^{u,\tau-1})^2 h^{u,\tau-1},\tag{11a}$$

$$g_{\text{DR}}^{\tau} = \bar{g}_{\text{DR}}n^{u,\tau-1},\tag{11b}$$

$$g_{\text{Ca}}^{\tau} = \bar{g}_{\text{Ca}}(s^{u,\tau-1})^2 z^{u,\tau-1},\tag{11c}$$

$$g_{\text{C}}^{\tau} = \bar{g}_{\text{C}}c^{u,\tau-1}\chi([\text{Ca}^{2+}]_{\text{nd}}^{u,\tau-1}),\tag{11d}$$

$$g_{\text{AHP}}^{\tau} = \bar{g}_{\text{AHP}}q^{u,\tau-1},\tag{11e}$$

where $\bar{g}_{\text{Na}}$, $\bar{g}_{\text{DR}}$, $\bar{g}_{\text{Ca}}$, $\bar{g}_{\text{C}}$, and $\bar{g}_{\text{AHP}}$ (S/cm$^2$) are the maximum ion conductances, $\chi$ (unitless) is the following function:

$$\chi([\text{Ca}^{2+}]_{\text{nd}}^{u,\tau}) = \min\left(\frac{v_{\text{Ca,n}}[\text{Ca}^{2+}]_{\text{nd}}^{u,\tau} - 99.8\cdot 10^{-6}}{2.5\cdot 10^{-4}},1\right),\tag{12}$$

and the gating variables $h^{u,\tau}$, $n^{u,\tau}$, $s^{u,\tau}$, $c^{u,\tau}$, $q^{u,\tau}$, $z^{u,\tau}$, and $m_{\infty}^{u,\tau}$ (unitless) are given by:

$$\frac{x^{u,\tau} - x^{u,\tau-1}}{\Delta t} = \alpha_{\text{x}}^{u,\tau}(1 - x^{u,\tau}) - \beta_{\text{x}}^{u,\tau}x^{u,\tau}, \quad \text{with } x \in \{h,n,s,c,q\},\tag{13a}$$

$$\frac{z^{u,\tau} - z^{u,\tau-1}}{\Delta t} = \frac{z_{\infty}^{u,\tau} - z^{u,\tau}}{\tau_{\text{z}}},\tag{13b}$$

$$m_{\infty}^{u,\tau} = \frac{\alpha_{\text{m}}^{u,\tau}}{\alpha_{\text{m}}^{u,\tau} + \beta_{\text{m}}^{u,\tau}},\tag{13c}$$

with

$$\alpha_{\mathrm{m}}^{u,\tau} = -\frac{3.2 \cdot 10^2 \cdot \phi_1}{\exp(-\phi_1/4) - 1}, \ \text{with} \ \phi_1 = \phi_{\mathrm{m,ns}}^{u,\tau} + 46.9 \tag{14a}$$

$$\beta_{\mathrm{m}}^{u,\tau} = \frac{2.8 \cdot 10^2 \cdot \phi_2}{\exp(\phi_2/5) - 1}, \ \text{with} \ \phi_2 = \phi_{\mathrm{m,ns}}^{u,\tau} + 19.9 \tag{14b}$$

$$\alpha_{\mathrm{h}}^{u,\tau} = 128 \exp\left(\frac{-43 - \phi_{\mathrm{m,ns}}^{u,\tau}}{18}\right), \tag{14c}$$

$$\beta_{\mathrm{h}}^{u,\tau} = \frac{4000}{1 + \exp(-\phi_3/5)}, \ \text{with} \ \phi_3 = \phi_{\mathrm{m,ns}}^{u,\tau} + 20 \tag{14d}$$

$$\alpha_{\mathrm{n}}^{u,\tau} = -\frac{16 \cdot \phi_4}{\exp(-\phi_4/5) - 1}, \ \text{with} \ \phi_4 = \phi_{\mathrm{m,ns}}^{u,\tau} + 24.9 \tag{14e}$$

$$\beta_{\mathrm{n}}^{u,\tau} = 250 \exp(-\phi_5/40), \ \text{with} \ \phi_5 = \phi_{\mathrm{m,ns}}^{u,\tau} + 40 \tag{14f}$$

$$\alpha_{\mathrm{s}}^{u,\tau} = \frac{1600}{1 + \exp(-0.072(\phi_{\mathrm{m,nd}}^{u,\tau} - 5))}, \tag{14g}$$

$$\beta_{\mathrm{s}}^{u,\tau} = \frac{20 \cdot \phi_6}{\exp(\phi_6/5) - 1}, \ \text{with} \ \phi_6 = \phi_{\mathrm{m,nd}}^{u,\tau} + 8.9 \tag{14h}$$

$$z_{\infty}^{u,\tau} = \frac{1}{1 + \exp(\phi_7)}, \ \text{with} \ \phi_7 = \phi_{\mathrm{m,nd}}^{u,\tau} + 30 \tag{14i}$$

$$\tau_z = 1, \tag{14j}$$

$$\alpha_{\mathrm{c}}^{u,\tau} = \begin{cases} 52.7 \exp\left(\dfrac{\phi_8}{11} - \dfrac{\phi_9}{27}\right), & \text{if} \ \phi_{\mathrm{m,nd}}^{u,\tau} \leq -10 \ \mathrm{mV} \\[2mm] 2000 \exp(-\phi_9/27), & \text{otherwise} \end{cases} \tag{14k}$$
$$\text{with} \ \phi_8 = \phi_{\mathrm{m,nd}}^{u,\tau} + 50 \ \text{and} \ \phi_9 = \phi_{\mathrm{m,nd}}^{u,\tau} + 53.5$$

$$\beta_{\mathrm{c}}^{u,\tau} = \begin{cases} 2000 \exp(-\phi_9/27) - \alpha_{\mathrm{c}}^{u,\tau}, & \text{if} \ \phi_{\mathrm{m,nd}}^{u,\tau} \leq -10 \ \mathrm{mV} \\[2mm] 0, & \text{otherwise} \end{cases} \tag{14l}$$

$$\alpha_{\mathrm{q}}^{u,\tau} = \min(2 \cdot 10^4 (v_{\mathrm{Ca,n}}[\mathrm{Ca}^{2+}]_{\mathrm{nd}}^{u,\tau} - 99.8 \cdot 10^{-6}), 10), \tag{14m}$$

$$\beta_{\mathrm{q}}^{u,\tau} = 1. \tag{14n}$$

In Eq 14, rates ($\alpha$'s and $\beta$'s) are in units of 1/s and $\tau_z$ is in units of s.

Further, all neuronal compartments contain Na$^+$, K$^+$, and Cl$^-$ leak channels, a 3Na$^+$/2K$^+$ pump, a K$^+$/Cl$^-$ cotransporter (KCC2), a Na$^+$/K$^+$/2Cl$^-$ cotransporter (NKCC1), and a

$2\mathrm{Na}^+/\mathrm{Ca}^{2+}$ exchanger. Their fluxes are given by:

$$j_{\mathrm{leak},k,\mathrm{n}l}^{u,\tau} = \bar{g}_{\mathrm{leak},k,\mathrm{n}l}(\phi_{\mathrm{m},\mathrm{n}l}^{u,\tau} - E_{k,\mathrm{n}l}^{u,\tau})/(Fz_k), \tag{15a}$$

$$j_{\mathrm{pump},\mathrm{n}l}^{u,\tau} = \frac{\rho_{\mathrm{n}}}{1.0 + \exp((25 - [\mathrm{Na}^+]_{\mathrm{n}l}^{u,\tau-1})/3)} \cdot \frac{1.0}{1.0 + \exp(3.5 - [\mathrm{K}^+]_{\mathrm{e}l}^{u,\tau-1})}, \tag{15b}$$

$$j_{\mathrm{kcc2},\mathrm{n}l}^{u,\tau} = U_{\mathrm{kcc2}} \ln\left(\frac{[\mathrm{K}^+]_{\mathrm{n}l}^{u,\tau-1}[\mathrm{Cl}^-]_{\mathrm{n}l}^{u,\tau-1}}{[\mathrm{K}^+]_{\mathrm{e}l}^{u,\tau-1}[\mathrm{Cl}^-]_{\mathrm{e}l}^{u,\tau-1}}\right), \tag{15c}$$

$$\begin{aligned}
j_{\mathrm{nkcc1},\mathrm{n}l}^{u,\tau} &= U_{\mathrm{nkcc1}} \frac{1}{1 + \exp(16 - [\mathrm{K}^+]_{\mathrm{e}l}^{u,\tau-1})} \\
&\cdot \left(\ln\left(\frac{[\mathrm{K}^+]_{\mathrm{n}l}^{u,\tau-1}[\mathrm{Cl}^-]_{\mathrm{n}l}^{u,\tau-1}}{[\mathrm{K}^+]_{\mathrm{e}l}^{u,\tau-1}[\mathrm{Cl}^-]_{\mathrm{e}l}^{u,\tau-1}}\right) + \ln\left(\frac{[\mathrm{Na}^+]_{\mathrm{n}l}^{u,\tau-1}[\mathrm{Cl}^-]_{\mathrm{n}l}^{u,\tau-1}}{[\mathrm{Na}^+]_{\mathrm{e}l}^{u,\tau-1}[\mathrm{Cl}^-]_{\mathrm{e}l}^{u,\tau-1}}\right)\right),
\end{aligned} \tag{15d}$$

$$j_{\mathrm{Ca-dec},\mathrm{n}l}^{u,\tau} = U_{\mathrm{Ca-dec}} \cdot ([\mathrm{Ca}^{2+}]_{\mathrm{n}l}^{u,\tau-1} - [\mathrm{Ca}^{2+}]_{\mathrm{basal},\mathrm{n}l}^{u}) \cdot \frac{\alpha_{\mathrm{n}l}^{\tau-1}}{\gamma_{\mathrm{m}}}, \tag{15e}$$

where $g_{\mathrm{leak},k,\mathrm{n}l}$ (S/cm$^2$) is the leak ion conductance, $\rho_{\mathrm{n}}$ (mMcm/s), $U_{\mathrm{kcc2}}$ (mMcm/s), and $U_{\mathrm{nkcc1}}$ (mMcm/s) are pump and cotransporter strengths, $U_{\mathrm{Ca-dec}}$ (s$^{-1}$) is the Ca$^{2+}$ decay rate, and $[\mathrm{Ca}^{2+}]_{\mathrm{basal},\mathrm{n}l}^{u}$ (mM) is the basal Ca$^{2+}$ concentration. Note that we treat the active ion channels and leak currents implicitly, except with respect to the gating variables, and the pumps and cotransporters explicitly.

### Glial membrane mechanisms

We implement the same glial membrane mechanisms as those used in Sætra et al. 2021 [28]. These include Na$^+$ and Cl$^-$ leak channels, an inward rectifying K$^+$ channel (Kir), and a 3Na$^+$/2K$^+$ pump in both layers:

$$j_{\mathrm{leak},k,\mathrm{g}l}^{u,\tau} = \bar{g}_{\mathrm{leak},k,\mathrm{g}l}(\phi_{\mathrm{m},\mathrm{g}l}^{u,\tau} - E_{k,\mathrm{g}l}^{u,\tau})/(Fz_k), \tag{16a}$$

$$j_{\mathrm{Kir}}^{u,\tau} = \bar{g}_{\mathrm{Kir}} f_{\mathrm{Kir}}(\phi_{\mathrm{m},\mathrm{g}l}^{u,\tau} - E_{\mathrm{K},\mathrm{g}l}^{u,\tau})/(Fz_{\mathrm{K}}), \tag{16b}$$

$$\begin{aligned}
f_{\mathrm{Kir}} &= \sqrt{\frac{[\mathrm{K}^+]_{\mathrm{e}l}^{u,\tau-1}}{[\mathrm{K}^+]_{\mathrm{basal},\mathrm{e}l}^{u}}} \left(\frac{1 + \exp(18.4/42.4)}{1 + \exp((\Delta\phi + 18.5)/42.5)}\right) \\
&\cdot \left(\frac{1 + \exp(-(118.6 + E_{\mathrm{basal},\mathrm{K},\mathrm{g}l}^{u})/44.1)}{1 + \exp(-(118.6 + \phi_{\mathrm{mg},l}^{u,\tau-1})/44.1)}\right)
\end{aligned} \tag{16c}$$

$$\begin{aligned}
j_{\mathrm{pump},\mathrm{g}l}^{u,\tau} &= \rho_{\mathrm{g}} \left(\frac{([\mathrm{Na}^+]_{\mathrm{g}l}^{u,\tau-1})^{1.5}}{([\mathrm{Na}^+]_{\mathrm{g}l}^{u,\tau-1})^{1.5} + ([\mathrm{Na}^+]_{\mathrm{threshold},\mathrm{g}})^{1.5}}\right) \\
&\cdot \left(\frac{[\mathrm{K}^+]_{\mathrm{e}l}^{u,\tau-1}}{[\mathrm{K}^+]_{\mathrm{e}l}^{u,\tau-1} + [\mathrm{K}^+]_{\mathrm{threshold},\mathrm{e}}}\right).
\end{aligned} \tag{16d}$$

Here, $\bar{g}_{\text{leak},k,gl}$ (S/cm$^2$) and $\bar{g}_{\text{Kir}}$ (S/cm$^2$) are ion conductantes, $[\text{K}^+]^u_{\text{basal},el}$ (mM) is the basal ECS K$^+$ concentration, $\Delta\phi = \phi^{u,\tau-1}_{\text{m},gl} - E^{u,\tau-1}_{\text{K},gl}$, $E^u_{\text{basal,K},gl}$ (mV) is the basal K$^+$ reversal potential, $\rho_{\text{g}}$ (mMcm/s) is the glial pump strength, and $[\text{Na}^+]_{\text{threshold,g}}$ (mM) and $[\text{K}^+]_{\text{threshold,e}}$ (mM) are the pump's threshold concentrations for Na$^+$ and K$^+$, respectively. The reversal potentials $E^{u,\tau}_{k,gl}$ are calculated as in Eq 10. Note that we treat the ion channels implicitly, except with respect to the Kir conductance, and the pump explicitly.

## Synaptic flux density

When we apply synaptic connections, we follow the same approach as in Sætra et al. 2021 [28]. The synaptic flux $j^{u'\to u,\tau}_{\text{syn}}$ flows across the dendritic membrane of the postsynaptic neuron in unit $u$, caused by neurotransmitter release of the presynaptic neuron in unit $u'$. The flux is excitatory and divided into ion-specific components where only the cations have non-zero contributions:

$$j^{u'\to u,\tau}_{\text{syn},k} = g^{u'\to u,\tau}_{\text{syn},k}(\phi^{u,\tau}_{\text{m,nd}} - E^{u,\tau}_{k,\text{nd}})/(Fz_k). \tag{17}$$

The conductance $g^{u'\to u,\tau}_{\text{syn},k}$ (S/cm$^2$) is given by a dual exponential function:

$$\begin{aligned} g^{u'\to u,\tau}_{\text{syn},k} &= S\bar{g}_{\text{syn},k}\sum_s\left[\exp\left(-\frac{t-(t^{n'}_{\text{AP},s}+t_{\text{delay}})}{\tau_1}\right) - \exp\left(-\frac{t-(t^{n'}_{\text{AP},s}+t_{\text{delay}})}{\tau_2}\right)\right] \\ &\cdot\Theta(t-(t^{n'}_{\text{AP}}+t_{\text{delay}})). \end{aligned} \tag{18}$$

Here, $S$ (unitless) is a synaptic strength factor, $\bar{g}_{\text{syn},k}$ (S/cm$^2$) is the maximum ion conductance, $\Theta(t)$ is the Heaviside function, and $\tau_1$ and $\tau_2$ are time constants. The $s$th spike-time of the presynaptic neuron is denoted $t^{n'}_{\text{AP},s}$, while $t_{\text{delay}}$ is the delay due to propagation of the action potential down its axon to the synaptic site.

We determine the spike times using a spiking threshold of −20 mV. When the membrane potential reaches the threshold, the spike time $t^{n'}_{\text{AP},s}$ is stored and the neuron is marked as currently spiking. The mark is removed when the membrane potential re-polarizes to −40 mV.

## Constant stimulus flux density

When applying constant stimulus currents, we adopt the same approach as outlined in Sætra et al. 2020 [1]. Specifically, we apply a transmembrane, inward K$^+$ current to the neuronal soma(s) of interest:

$$j^{u,\tau}_{\text{stim}} = \bar{j}_{\text{stim}}. \tag{19}$$

Here, $\bar{j}_{\text{stim}}$ (mMcm/s) is constant.

### Total membrane flux densities

We summarize the total membrane flux densities here for easy reference:

$$j_{m,Na,ns}^{u,\tau} = j_{Na}^{u,\tau} + j_{leak,Na,ns}^{u,\tau} + 3j_{pump,ns}^{u,\tau} + j_{nkcc1,ns}^{u,\tau} - 2j_{Ca-dec,ns}^{u,\tau}, \tag{20a}$$

$$j_{m,K,ns}^{u,\tau} = j_{DR}^{u,\tau} + j_{leak,K,ns}^{u,\tau} - 2j_{pump,ns}^{u,\tau} + j_{nkcc1,ns}^{u,\tau} + j_{kcc2,ns}^{u,\tau} + j_{stim}^{u,\tau}, \tag{20b}$$

$$j_{m,Cl,ns}^{u,\tau} = j_{leak,Cl,ns}^{u,\tau} + 2j_{nkcc1,ns}^{u,\tau} + j_{kcc2,ns}^{u,\tau}, \tag{20c}$$

$$j_{m,Ca,ns}^{u,\tau} = j_{Ca-dec,ns}^{u,\tau}, \tag{20d}$$

$$j_{m,Na,nd}^{u,\tau} = j_{leak,Na,nd}^{u,\tau} + 3j_{pump,nd}^{u,\tau} + j_{nkcc1,nd}^{u,\tau} - 2j_{Ca-dec,nd}^{u,\tau} + j_{syn,Na}^{u'\to u,\tau}, \tag{20e}$$

$$j_{m,K,nd}^{u,\tau} = j_{AHP}^{u,\tau} + j_{C}^{u,\tau} + j_{leak,K,nd}^{u,\tau} - 2j_{pump,nd}^{u,\tau} + j_{nkcc1,nd}^{u,\tau} + j_{kcc2,nd}^{u,\tau} + j_{syn,K}^{u'\to u,\tau}, \tag{20f}$$

$$j_{m,Cl,nd}^{u,\tau} = j_{leak,Cl,nd}^{u,\tau} + 2j_{nkcc1,nd}^{u,\tau} + j_{kcc2,nd}^{u,\tau}, \tag{20g}$$

$$j_{m,Ca,nd}^{u,\tau} = j_{Ca}^{u,\tau} + j_{Ca-dec,nd}^{u,\tau} + j_{syn,Ca}^{u'\to u,\tau}, \tag{20h}$$

$$j_{m,Na,gs}^{u,\tau} = j_{leak,Na,gs}^{u,\tau} + 3j_{pump,gs}^{u,\tau}, \tag{20i}$$

$$j_{m,K,gs}^{u,\tau} = j_{Kir,gs}^{u,\tau} - 2j_{pump,gs}^{u,\tau}, \tag{20j}$$

$$j_{m,Cl,gs}^{u,\tau} = j_{leak,Cl,gs}^{u,\tau}, \tag{20k}$$

$$j_{m,Na,gd}^{u,\tau} = j_{leak,Na,gd}^{u,\tau} + 3j_{pump,gd}^{u,\tau}, \tag{20l}$$

$$j_{m,K,gd}^{u,\tau} = j_{Kir,gd}^{u,\tau} - 2j_{pump,gd}^{u,\tau}, \tag{20m}$$

$$j_{m,Cl,gd}^{u,\tau} = j_{leak,Cl,gd}^{u,\tau}. \tag{20n}$$

### Boundary conditions

We apply either sealed-end boundary conditions, that is, no ions are allowed to enter or leave the system, or periodic boundary conditions, that is, ions are allowed to move between unit 1 and unit $U$.

### Initial conditions

The initial conditions (including ion concentrations, gating variables, and intracellular volume fractions) and baseline values (including extracellular volume fractions and membrane potentials) are listed in Table 3. We used the same initial values for all units, regardless of their number. To set the initial conditions, we used the values from Sætra et al. 2021 [28] as a starting point. We then calibrated the model by simulating a single unit free from stimulus until it reached steady-state.

**Table 3. Initial conditions and baseline values.** Values with more decimals included were used in the simulations and are available with the source code.

| Variable | Value | Variable | Value |
|---|---|---|---|
| $[\text{Na}^+]_{nl}^{u,0}$ | 18.7 mM | $h^{u,0}$ | 0.9993 |
| $[\text{Na}^+]_{gl}^{u,0}$ | 14.5 mM | $n^{u,0}$ | 0.0003 |
| $[\text{Na}^+]_{el}^{u,0}$ | 142.2 mM | $s^{u,0}$ | 0.0076 |
| $[\text{K}^+]_{nl}^{u,0}$ | 138.1 mM | $c^{u,0}$ | 0.0056 |
| $[\text{K}^+]_{gl}^{u,0}$ | 101.2 mM | $q^{u,0}$ | 0.0117 |
| $[\text{K}^+]_{el}^{u,0}$ | 3.54 mM | $z^{u,0}$ | 1.0 |
| $[\text{Cl}^-]_{nl}^{u,0}$ | 7.14 mM | $\alpha_{nl}^{u,0}$ | 0.4 |
| $[\text{Cl}^-]_{gl}^{u,0}$ | 5.67 mM | $\alpha_{gl}^{u,0}$ | 0.4 |
| $[\text{Cl}^-]_{el}^{u,0}$ | 131.9 mM | $\alpha_{el}^{u,0}$ | 0.2 |
| $[\text{Ca}^{2+}]_{nl}^{u,0}$ | 0.01 mM | $\phi_{\text{m},nl}^{u,0}$ | −67 mV |
| $[\text{Ca}^{2+}]_{el}^{u,0}$ | 1.1 mM | $\phi_{\text{m},gl}^{u,0}$ | −84 mV |

To ensure electroneutrality of the system, we define a set of immobile ions $a_{rl}^u$ that we calculate at the beginning of each simulation:

$$a_{rl}^u = \frac{\gamma_\text{m} C_\text{m}}{z_0 F}\phi_{\text{m},rl}^{u,0} - \frac{\alpha_{rl}^{u,0}}{z_0}\sum_k z_k [k]_{rl}^{u,0}, \quad r = \text{n}, \text{g} \tag{21a}$$

$$a_{el}^u = -\frac{\gamma_\text{m} C_\text{m}}{z_0 F}\left(\phi_{\text{m},nl}^{u,0} + \phi_{\text{m},gl}^{u,0}\right) - \frac{\alpha_{el}^{u,\tau}}{z_0}\sum_k z_k [k]_{el}^{u,0}. \tag{21b}$$

Here, $z_0$ denotes the valency of the immobile ions, which we set to −1. Note that since $a_{rl}^u$ are constant, they are given as moles per total volume.

## Simulation details

A summary of the different simulation protocols used in this paper is given in Table 4, including approximate simulation runtimes. Note that we utilized the Numba library [51] to speed up computations, and the reported runtimes do not include ahead-of-time compilation times. Timings were conducted on a Lenovo ThinkPad X1 Carbon Gen 8 with an Intel Core i7–10510U CPU at 1.80 GHz and Ubuntu 22.04.4 LTS.

## Numerical implementation

Each time step $\tau \Delta t$ constitutes two substeps:

1. Find $\alpha_{\{\text{n,g}\}l}^{u,\tau}$, $[k]_{rl}^{u,\tau}$, and $\phi_{rl}^{u,\tau}$ using Newton's method and the gating variables from the previous time step.

2. Update the gating variables using Eq 13 and $[k]_{rl}^{u,\tau}$, $\phi_{rl}^{u,\tau}$, and $\alpha_{\{\text{n,g}\}l}^{u,\tau}$ from substep 1.

We implemented the model in Python 3.10, using a time step of $\Delta t = 5 \cdot 10^{-5}$ s. The source code can be downloaded from https://github.com/martejulie/electrodiffusive-network-model.

**Table 4. Simulation protocols.**

| Protocol property | Figs 2–4 | Figs 5 and 6 † | Figs 7 and 8 | Figs 9–11 * |
|---|---|---|---|---|
| Number of units, $U$ | 10 | 1 | 25 | 2 |
| Distance between units, $\Delta x_u$ | 5 μm | – | 1 μm | 1 mm |
| Boundary conditions | Periodic | Closed | Closed | Closed |
| Synaptic connections | Yes | – | No | No |
| External stimulus | 1 spike (Unit 1) | Spike train¤ | Constant (Units 1–15) | Constant (Unit 1) |
| Synaptic delay, $t_{delay}$ | 10 ms | 0 ms | – | – |
| Synaptic strength factor, $S$ | 14 | 1 | – | – |
| $\bar{j}_{stim}$ | – | – | $5 \cdot 10^{-4}$ mMcm/s | $2 \cdot 10^{-4}$ mMcm/s |
| Stimulus onset | 0.1 s | 0.1 s | 0.1 s | 0.1 s |
| Stimulus offset | – | 10 s | 10 s | 10 s |
| Simulation duration | 1.5 s | 10 s | 10 s | 10 s |
| Simulation runtime | 5 min | 7 min | 25 min | 7 min |

†For the reduced model, we ignored ion concentration dynamics by setting all reversal potentials, pumps, and cotransporters constant (using initial ion concentrations for the soma), except for the 2Na$^+$/Ca$^{2+}$ exchanger. We allowed Ca$^{2+}$ to dynamically affect the conductance of Ca$^{2+}$-sensitive ion channels.

*For the strong-glia model, we multiplied the glial interunits fluxes $j_{interunits,k,gl}^{l,\tau}$ by a factor 2000.

¤Spike times were given by a 400 Hz Poisson spike train, which we generated using the `homogeneous_poisson_process` function from the Elephant software package [52]. We saved the spike train to file to ensure that the full- and reduced models received identical spike trains.

## Supporting information

**S1 Fig. Homeostatic recovery of the system.** Temporal evolution of the neuronal membrane potential (**A**) and change in ECS K$^+$ concentration (**B**), neuronal K$^+$ concentration (**C**), glial K$^+$ concentration (**D**), and ECS volume fraction (**E**) in the soma layer of unit 1. The figure is based on the simulation presented in Fig 2, except all synaptic currents were turned off at $t = 1.5$ s.
(PDF)

**S2 Fig. Volume dynamics in pathological conditions.** Temporal evolution of the neuronal (**A**), glial (**B**), and extracellular (**C**) volume fractions of the somatic layer in unit 1, given as change from baseline values. The figure is based on the simulation presented in Fig 7.
(PDF)

## Acknowledgments

MJS and YM would like to thank Geir Halnes, Ada J. Ellingsrud, and Marie E. Rognes for useful discussions during the initial phase of the project. MJS and YM acknowledge the use of ChatGPT-3 (OpenAI) to proofread the manuscript and identify improvements in the writing style.

## Author Contributions

**Conceptualization:** Marte J. Sætra, Yoichiro Mori.

**Funding acquisition:** Marte J. Sætra, Yoichiro Mori.

**Investigation:** Marte J. Sætra.

**Methodology:** Marte J. Sætra, Yoichiro Mori.

**Software:** Marte J. Sætra.

**Supervision:** Yoichiro Mori.

**Validation:** Marte J. Sætra.

**Visualization:** Marte J. Sætra.

**Writing – original draft:** Marte J. Sætra.

**Writing – review & editing:** Marte J. Sætra, Yoichiro Mori.

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
