## [Decision Letter · Decision Letter 0]

18 Jun 2024

Dear Dr. Sætra,

Thank you very much for submitting your manuscript "An electrodiffusive network model with multicompartmental neurons and synaptic connections" for consideration at PLOS Computational Biology.

As with all papers reviewed by the journal, your manuscript was reviewed by members of the editorial board and by several independent reviewers. In light of the reviews (below this email), we would like to invite the resubmission of a significantly-revised version that takes into account the reviewers' comments.

in particular, you need to show the results for a model that lacks drift in order to demonstrate the role of drift.  In addition, you will need to do additional simulations demonstrating robustness (or sensitivity) to parameter values.

We cannot make any decision about publication until we have seen the revised manuscript and your response to the reviewers' comments. Your revised manuscript is also likely to be sent to reviewers for further evaluation.

Sincerely,

Kim T. Blackwell, V.M.D., Ph.D.

Academic Editor

PLOS Computational Biology

Daniele Marinazzo

Section Editor

PLOS Computational Biology

Reviewer's Responses to Questions

**Comments to the Authors:**

Reviewer #1: The paper presents an intriguing extension of the author's prior work, utilizing the Kirchhoff-Nernst-Planck framework to model ion electrodiffusion within a two-compartment neuron, accompanied by glial cells and the extracellular space. They augment this by incorporating synaptic coupling in a classic ring model formation and explore pathological conditions, demonstrating how excessive stimulation can induce depolarization block, a phenomenon they then demonstrate is ameliorated by sufficient astrocytic clearance of extracellular potassium.

I recommend the paper for publication, but have a few queries and suggestions for improvement:

Figure Presentation: It is quite difficult to follow the individual traces. As the plot appears twice, could you show/highlight one example in either Fig2B or Fig3A.

Resting Membrane Potential (RMP) Dynamics: It seems that the RMP decreases between each action potential in Fig 2B. Given the expected increase in the Nernst potential for K+ (with a relatively larger increase in extracellular concentration than the reduction in neuronal concentration), is the change in RMP primarily driven by Na+ dynamics?

Extracellular Volume Fraction Changes: While the change in volume is minor, increased neuronal activity typically results in reduced extracellular volume. Have you observed a reduction in ECS volume in pathological simulations (Fig 7 & Fig 8)? Additionally, what drives the increase in extracellular volume fraction in Fig 4? If Na+ dynamics contribute, consider including them in a figure or discussing them in the text.

Homeostatic Recovery; Line 161 suggests an expected return to baseline values due to homeostatic activity. Could you clarify if there is any recovery over the short period between spikes, such as in the somatic neuronal K+ concentration (Fig 4E)? Have you experimented with halting spiking to observe if it returns to baseline or enters another stable state?

Calcium Flux: Considering the low concentration of calcium involved, displaying the calcium flux on a different scale in Fig 6A,B might enhance clarity.

Astrocyte K+ Buffering: Regarding the representation of inactive regions in the model, particularly when u=2, could the use of a 1D simulation exaggerate concentration changes due to volume of available tissue increase with the cube of the distance from the active region? While this may not impact the conclusion about potassium buffering preventing SD, it might be worth noting in the text for clarity.

ECS Size: The diagram in figure 10A suggests that the ECS size remains constant regardless of glial presence, but Equation (2) implies it should be larger without glia. Please clarify this in the caption.

Simulation details: You note in the discussion that a 2D or 3D model would be computationally demanding, I am curious how computational intensive these 1D simulations were. Could you add how long the simulations took to run in Table 4, or add comment on it in the text.

Reviewer #2: In this manuscript, the authors present a network level model of electrodiffusion suggesting that this model bypasses the need to consider neuronal geometry but can still predict ion concentrations. They develop a comprehensive multicompartment model and test different scenarios. However, how the simulation case studies or the motivation is biologically relevant is unclear as written. I have the following major concerns or questions:

1) Introduction: the motivation is given by the fact that even though many models exist, they cannot be used to study the spread of depression. But then I didn't find any evidence of what happens to ionic concentrations in depression or that this model has the potential to shed light on depression. The introduction teases at different important scenarios but doesn't clearly make the case for what scenarios (with experimental measurements) motivate their model. They also talk about computationally expensive spatial models but how expensive is expensive?

Typos: line 63 and line 35.

2) Figure 1: I had a very hard time understanding the compartments in Figure 1. I understand soma and dendrites for neurons. What does each unit represent? That every neuron is paired with a glial cell? are there no units with just neurons and ECS?

3) Parameter choices: The authors mention that they tune parameters manually (line 128) to obtain certain outcomes but I didn't see any mention of parametric sensitivity analysis?

4) The model has many free parameters, how we do know that any of the effects shown are robust?

5) Line 139: What is the functional consequence of drop in glial membrane voltage at t=1.5 s?

6) Results: This comment summarizes the overall concern I have with the paper. In figure 3C, the authors zoom in and show an extremely small difference and suggest that it is important. I have a very hard time understanding why differences that are practically negligible interesting and important. If one were to add noise to this system, would any of these outcomes hold? Later, there are various simulations of reduced models and comparison models. I am left wondering -- what is the point? The differences are so small that I don't know how to interpret any of the effects.

7) How is the reader to understand the different network configurations? I just really had a hard time building confidence in the biological importance of these findings.

Reviewer #3: The authors present the results from an electro-diffusive neuronal tissue model. The model incorporates compartments for neuronal, glial and extracellular spaces that they couple in multiple ways. The complexity of the neuronal models is moderate and the results are consistent with previous work. The incorporation of electrodiffusive dynamics is an important step forward for neuronal network models, but I feel that the manuscript needs to address a few small changes.

The assumptions in the physical model appear sound, but it would be good to provide more physiological motivation for the different models, i.e. no/weak/strong glial coupling. For instance what is the physiological interpretation of a diffusion length of 1 mm?

Aside from potential typos I may have missed, the model looks reasonable under the time scale/space scales involved here.

The mode is capable of producing ephaptic ionic and field effects. However it is not clear that the ionic part of this coupling is any different than the ionic coupling incorporated in many other network models.

The general outline of the KNP framework should be concisely reviewed/described in this paper. It is important to the understanding of the physics/physiology of the model.

Figure 7. Partially claims that the ephaptic coupling gives rise to the multiple action potentials, however they are gradually depolarizing up to this point and seem to just be reaching their thresholds within close temporal proximity.

The authors don't show the results for their model including diffusion, but lacking drift. The advancement of the model over previous models is the drift, so this comparison would help to elucidate the effects of the ephaptic field effects.

In general the effects of the ephaptic field are small, except in the extracellular drift for strong glia, and they are not highlighted by the authors. As the main advancement to the model, the authors could do more to give support for its necessity.

Figure 8. Heat maps don’t add much to the story.

Figure 9. Is there anything to see in the unit 2 voltage? If they don’t change, perhaps best to leave them off.

Line 518 “This approach guarantees conservation of ions.” This should be explained in the text. The reference doesn’t claim to conserve, only minimized disruption.

**Have the authors made all data and (if applicable) computational code underlying the findings in their manuscript fully available?**

Reviewer #1: Yes

Reviewer #2: Yes

Reviewer #3: Yes

PLOS authors have the option to publish the peer review history of their article (what does this mean?). If published, this will include your full peer review and any attached files.

Reviewer #1: No

Reviewer #2: No

Reviewer #3: No
---

## [Decision Letter · Decision Letter 1]

29 Oct 2024

Dear Dr. Sætra,

We are pleased to inform you that your manuscript 'An electrodiffusive network model with multicompartmental neurons and synaptic connections' has been provisionally accepted for publication in PLOS Computational Biology.

Best regards,

Kim T. Blackwell, V.M.D., Ph.D.

Academic Editor

PLOS Computational Biology

Daniele Marinazzo

Section Editor

PLOS Computational Biology

Feilim Mac Gabhann

Editor-in-Chief

PLOS Computational Biology

Jason Papin

Editor-in-Chief

PLOS Computational Biology

Reviewer's Responses to Questions

**Comments to the Authors:**

Reviewer #1: Thank you for the thorough responses to all my questions, I particularly appreciated the additional supplementary figures.

Reviewer #2: I appreciate the authors writing out detailed responses to my comments. In fact, I found the response to reviewer’s illustrative and of great clarity, probably much more so than the manuscript itself. I am mostly satisfied with the response. I think the issue of what does this mean for biology still remains as the authors admit themselves.

**Have the authors made all data and (if applicable) computational code underlying the findings in their manuscript fully available?**

Reviewer #1: Yes

Reviewer #2: Yes

PLOS authors have the option to publish the peer review history of their article (what does this mean?). If published, this will include your full peer review and any attached files.

Reviewer #1: No

Reviewer #2: No

---

## [Editor Report · Acceptance letter]

4 Nov 2024

PCOMPBIOL-D-24-00707R1 

An electrodiffusive network model with multicompartmental neurons and synaptic connections

Dear Dr Sætra,

I am pleased to inform you that your manuscript has been formally accepted for publication in PLOS Computational Biology. Your manuscript is now with our production department and you will be notified of the publication date in due course.

With kind regards,

Anita Estes
